# Clarinet (CLA-1), a novel active zone protein required for synaptic vesicle clustering and release

Zhao Xuan[1,2,3†], Laura Manning[4†], Jessica Nelson[1,2,3], Janet E Richmond[4], Daniel A Colón-Ramos[1,2,3,5], Kang Shen[6,7], Peri T Kurshan[6*]

[1]Program in Cellular Neuroscience, Neurodegeneration and Repair, Yale University School of Medicine, New Haven, United States; [2]Department of Cell Biology, Yale University School of Medicine, New Haven, United States; [3]Department of Neuroscience, Yale University School of Medicine, New Haven, United States; [4]Department of Biological Sciences, University of Illinois at Chicago, Chicago, Illinois; [5]Instituto de Neurobiología, Recinto de Ciencias Médicas, Universidad de Puerto Rico, San Juan, Puerto Rico; [6]Department of Biology, Stanford University, Stanford, United States; [7]Howard Hughes Medical Institute

*For correspondence:
pkurshan@stanford.edu

†These authors contributed equally to this work

**Abstract** Active zone proteins cluster synaptic vesicles at presynaptic terminals and coordinate their release. In forward genetic screens, we isolated a novel *Caenorhabditis elegans* active zone gene, *clarinet* (*cla-1*). *cla-1* mutants exhibit defects in synaptic vesicle clustering, active zone structure and synapse number. As a result, they have reduced spontaneous vesicle release and increased synaptic depression. *cla-1* mutants show defects in vesicle distribution near the presynaptic dense projection, with fewer undocked vesicles contacting the dense projection and more docked vesicles at the plasma membrane. *cla-1* encodes three isoforms containing common C-terminal PDZ and C2 domains with homology to vertebrate active zone proteins Piccolo and RIM. The C-termini of all isoforms localize to the active zone. Specific loss of the ~9000 amino acid long isoform results in vesicle clustering defects and increased synaptic depression. Our data indicate that specific isoforms *of clarinet* serve distinct functions, regulating synapse development, vesicle clustering and release.
DOI: https://doi.org/10.7554/eLife.29276.001

## Introduction

The coordinated and precise release of synaptic vesicles from presynaptic compartments underlies neuronal communication and brain function. This is achieved through the concerted action of conserved proteins that make up the cytomatrix at the active zone, a protein dense region within the presynaptic bouton that is surrounded by synaptic vesicles. Active zone proteins regulate neurotransmission by recruiting synaptic vesicles to the plasma membrane, positioning calcium channels adjacent to the site of exocytosis, and priming synaptic vesicles for calcium-dependent release. In vertebrates, the main active zone proteins that coordinate synaptic vesicle release are Liprin-α, RIM, RIM-BP, Elks and Munc-13 (*Südhof, 2012*; *Ackermann et al., 2015*).

Two additional proteins, Bassoon and Piccolo, serve to cluster synaptic vesicles near the active zone (*Cases-Langhoff et al., 1996*; *Langnaese et al., 1996*; *Mukherjee et al., 2010*). Although the core components of the active zone are conserved between vertebrates and invertebrates, Bassoon and Piccolo have long been considered exclusive to vertebrates. While the N-terminus of Drosophila Bruchpilot (BRP) contains significant sequence homology to vertebrate ELKS (*Wagh et al., 2006*; *Kittel et al., 2006*), like Bassoon and Piccolo it also has a large C-terminal domain rich in coiled-coil

**eLife digest** Nerve cells, or neurons, communicate with one another by sending messages across junctions called synapses. When the neuron on one side of a synapse becomes active, calcium ions flood into the cell. This causes the neuron to release signals called neurotransmitters, which activate the cell on the other side of the synapse. Neurons store their neurotransmitter molecules inside packages called vesicles, and keep them clustered close to their release site, a region called the active zone. The active zone contains a number of different proteins. Some of these hold vesicles in position. Others respond to calcium entry by fusing vesicles with the cell membrane.

The identity of many of the proteins within the active zone remains unknown, especially those responsible for keeping vesicles clustered nearby. Xuan, Manning et al. therefore introduced mutations at random into the genome of the roundworm *C. elegans*, and searched for mutant worms with altered patterns of vesicles. Worms with mutations in a previously unknown gene, which they named *clarinet*, showed abnormal distribution of vesicles within the presynaptic compartment. They also released fewer vesicles compared to non-mutant worms.

Further experiments revealed that the *clarinet* gene encodes three different proteins with varying sizes, all found at the active zone. Using microscopy and electrode recordings, as well as a genetic technique called CRISPR, Xuan, Manning et al. showed that the three forms of clarinet have different roles. The shorter ones contribute to the development of synapses. They help ensure that the active zone forms correctly and that neurons have an appropriate number of synaptic connections. The longest form of clarinet is responsible for clustering vesicles, which allows cells to continue releasing vesicles during bursts of repeated neuronal firing.

Problems with synapses contribute to many brain disorders, including autism and intellectual disability. Xuan, Manning et al. hope that an increased understanding of how synapses form, and how they work, will provide insights into these and other conditions.

DOI: https://doi.org/10.7554/eLife.29276.002

structures that is thought to function in tethering synaptic vesicles (*Matkovic et al., 2013*). Recently, *Drosophila* Fife, a protein that contains ZnF, PDZ and C2 domains, was discovered based on sequence homology to the PDZ domain of vertebrate Piccolo, and shown to be an active zone protein (*Bruckner et al., 2012*). Fife binds to and functionally interacts with Rim to dock synaptic vesicles and increase probability of release (*Bruckner et al., 2017*). No clear homologs of Piccolo, Bassoon, Fife, or of the coiled-coil domain of BRP have been identified for *C. elegans*.

We performed forward genetic screens in *C. elegans* for proteins required for synaptic vesicle clustering, and identified clarinet (*cla-1*). CLA-1 is required for normal synapse number and *cla-1* null mutants exhibit reduced spontaneous synaptic vesicle release. *Cla-1* mutants have a smaller dense projection and display defects in the clustering of the active zone protein SYD-2/Liprin-α. They exhibit a dramatic reduction in the number of synaptic vesicles contacting the dense projection, and increased synaptic depression. The *cla-1* gene encodes three main isoforms (CLA-1L, CLA-1M and CLA-1S) containing PDZ and C2 domains with sequence homology to vertebrate Piccolo and RIM. While all three isoforms share a C-terminal region that localizes to the active zone, their genetic requirement in synapse function and development differ: the N-terminus of CLA-1L is specifically required for synaptic vesicle clustering and proper synaptic function during repeated stimulations, whereas the shorter isoforms or C-terminus are required for active zone assembly and proper synapse number. Together our findings indicate that *cla-1* encodes novel active zone proteins that are required for proper synapse development, active zone structure and synaptic vesicle clustering, and thus play a role in synaptic function during prolonged activation.

## Results

### CLA-1 is required in the NSM neuron for synaptic vesicle clustering

We performed unbiased forward genetic screens to identify molecules required for the localization of synaptic vesicle proteins in the serotonergic NSM neuron of the nematode *C. elegans* (*Figure 1A–C*). From this screen, we identified allele *ola104*, which displayed a diffuse distribution of

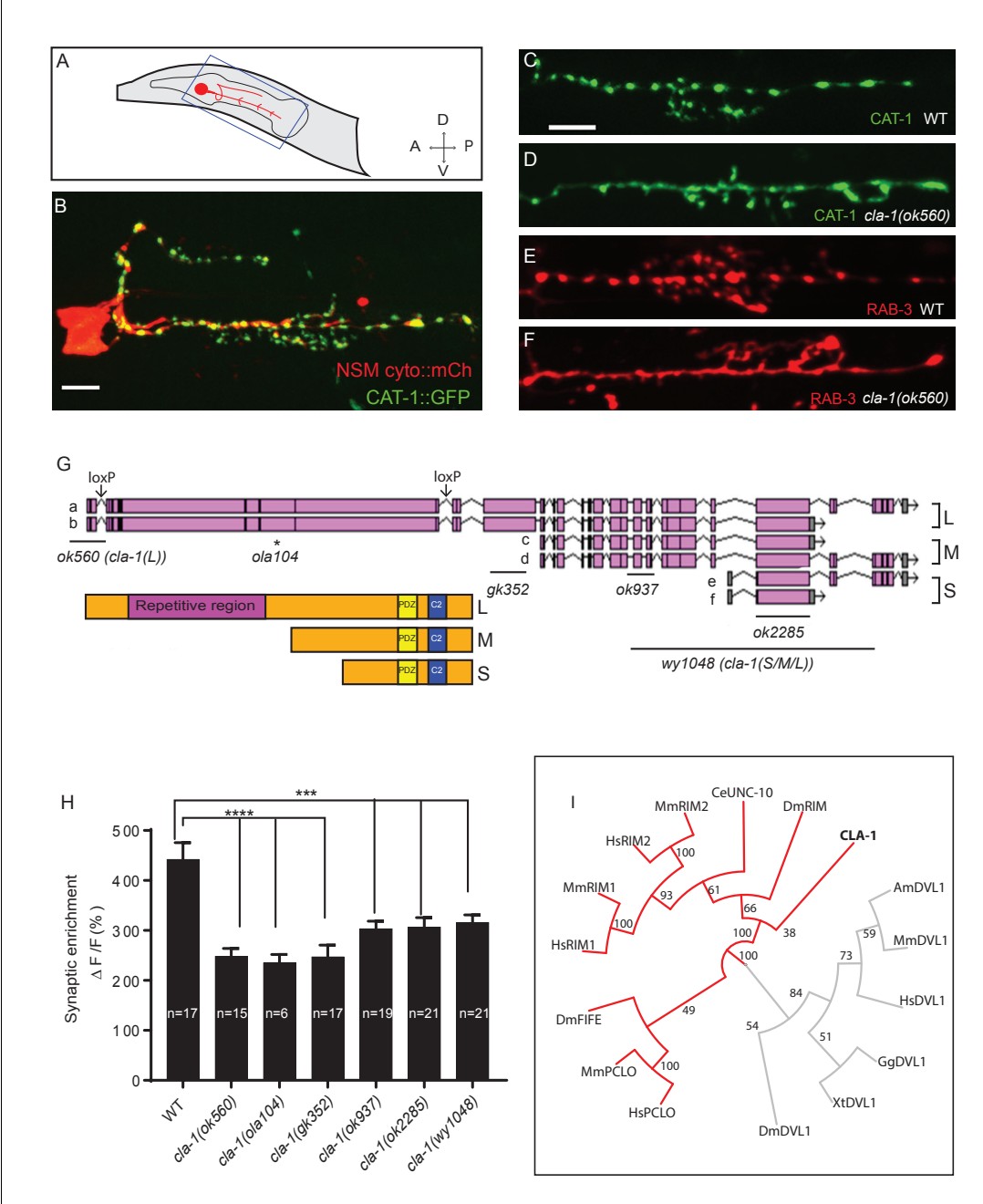

**Figure 1.** *cla-1* mutants display disrupted synaptic vesicle clustering in NSM neurons. (**A**) Schematic diagram of the nematode head and the NSM neuron (in red inside blue-boxed region). (**B**) Cytosolic mCherry (cyto::mCh) and the synaptic vesicle marker CAT-1::GFP expressed cell specifically in NSM. Scale bar = 5 μm. (**C–F**) Synaptic vesicle markers in NSM: CAT-1::GFP (**C–D**) or RAB-3::mCherry (**E–F**) in ventral neurite in wild type (WT; **C and E**) and *cla-1(ok560)* (**D and F**). Note how *cla-1* mutants exhibit diffuse (**D, F**) rather than the wild type punctate (**C, E**) fluorescence patterns. Scale bar = 5 μm. (**G**) Schematics of the genomic region of *cla-1* and the structure of three main isoforms of the CLA-1 protein. The locations of loxP sites and the genetic lesions of the *cla-1* alleles examined in this study are indicated. In addition to the common C-terminus, CLA-1L contains a large N-terminal repetitive region (see ***Figure 1—figure supplement 1G***). (**H**) Synaptic enrichment (ΔF/F) of CAT-1::GFP in NSM is greatly reduced in all *cla-1* mutants compared to wild type (WT). n = number of animals. (**I**) The PDZ sequence of CLA-1 was aligned to RIM, Piccolo and Fife from *C. elegans* (CeUNC-10), *Drosophila* (DmRIM, DmFife), mouse (MmRIM1/2, MmPCLO) and human (HsRIM1/2, HsPCLO) by neighbor joining with 100 bootstrap replicates. PDZ domains of Dishevelled family proteins were used as an outgroup (grey).

DOI: https://doi.org/10.7554/eLife.29276.003

The following figure supplements are available for figure 1:

**Figure supplement 1.** *ola104* displays disrupted synaptic vesicle clustering in NSM neuron and was identified as a genetic lesion of *cla-1*.

*Figure 1 continued on next page*

*Figure 1 continued*

DOI: https://doi.org/10.7554/eLife.29276.004

**Figure supplement 2.** Expression pattern of CLA-1 isoforms.

DOI: https://doi.org/10.7554/eLife.29276.005

the synaptic vesicle protein VMAT/CAT-1 as compared to wild type controls (*Figure 1—figure supplement 1A–D*). In *ola104* mutants, reduced intensity of synaptic puncta was accompanied by an increase in the extrasynaptic signal, suggestive of a defect in synaptic vesicle clustering at the synapse. Using single-nucleotide polymorphism mapping, we identified *ola104* as a missense mutation in *cla-1* (*Figure 1—figure supplement 1E*). An independent allele, *cla-1(ok560),* phenocopied and failed to complement *ola104* (*Figure 1C–F* and *Figure 1—figure supplement 1F*).

*cla-1* is predicted to encode six isoforms of different lengths (*Figure 1G*). Based on the length of the proteins, we classified them into three categories: CLA-1L (long) including CLA-1a and b; CLA-1M (medium) including CLA-1c and d; CLA-1S (short) including CLA-1e and f (*Figure 1G*). Distinct alleles affect different isoforms. *cla-1(ok560)* results in a deletion of the promoter and part of the coding region of cla-1L, and will be referred to henceforth as *cla-1(L). cla-1(wy1048)*, an allele we generated using CRISPR, eliminates most of cla-1S and M, including the PDZ and C2 domains. Because these domains are shared by all isoforms, this deletion is likely a null and the allele will henceforth be referred to as *cla-1(S/M/L)*. Importantly, in *cla-1(L)* deletion mutants, the shorter isoforms are still expressed, as evidenced by RT-PCR to the C-terminal PDZ domain (*Figure 1—figure supplement 1H*). Synaptic vesicle clustering was examined in five alleles affecting different isoforms (*Figure 1G*), and all alleles examined showed defects in synaptic vesicle clustering in NSM (*Figure 1H*). Since the long-isoform-specific allele *cla-1(L)* exhibited as dramatic a defect as the null allele, we hypothesize that CLA-1L may thus be specifically required for proper clustering of vesicles at the synapse.

## Structure, homology and expression pattern of CLA-1 isoforms

CLA-1L is composed of approximately 9000 amino acids and contains an extended repetitive region of about 4000 amino acids (*Figure 1G*). The 12 kb cDNA sequence encoding the repetitive region is comprised of tandem repeats, with a 282 bp repeat unit (*Figure 1—figure supplement 1G*). The secondary structure of the repetitive region is predicted to consist of random coils interlaced with alpha helices. CLA-1M is made up of ~3000 amino acids, whereas CLA-1S is ~1000 amino acids long. The common C-terminal domain for all three isoforms includes PDZ and C2 domains that are conserved with the mammalian active zone proteins Piccolo and RIM (*Figure 1I*). Other than the PDZ and C2 domains, we did not identify other sequence similarities between the CLA-1 isoforms and vertebrate sequences.

Based on a phylogenetic analysis using the PDZ domain sequences, we found that the *cla-1* PDZ domain is most similar to that of RIM, but constitutes a distinct clade (*Figure 1I*). This result, along with the lack of sequence homology between the rest of the CLA-1 protein (other than the C2 domains) and any known active zone proteins, suggests that *cla-1* encodes a novel active zone protein. Its role in synaptic vesicle clustering suggested that it may be functionally homologous to Piccolo, Bassoon and Fife, and hence was given the name Clarinet (CLA-1) to reflect its large size.

To determine the expression pattern of CLA-1 isoforms, we created GFP reporters under the *cla-1* promoters (2 kb fragments upstream of the L, M and S isoforms). We found that each isoform is expressed broadly within the nervous system, as evidenced by a high degree of colocalization with an mCherry reporter under the pan-neuronal *rab-3* promoter (*Figure 1—figure supplement 2A–C*). CLA-1S was expressed broadly throughout the nervous system, while CLA-1M and L were expressed in a subset of neurons.

To probe the subcellular localization of CLA-1L, we inserted GFP at the N-terminus of the endogenous *cla-1* locus via CRISPR (*Figure 2—figure supplement 1A*; *Dickinson et al., 2015*). Using this strain, we determined that GFP::CLA-1L (homozygous endogenous) localizes to synapses at the developmental period in which the embryonic nervous system begins to form (three-fold stage: *Figure 1—figure supplement 2D,E*). CLA-1L localized in a pattern reminiscent of synaptic vesicle marker RAB-3. When we expressed *mCherry::rab-3* cDNA under the NSM-specific promoter in the

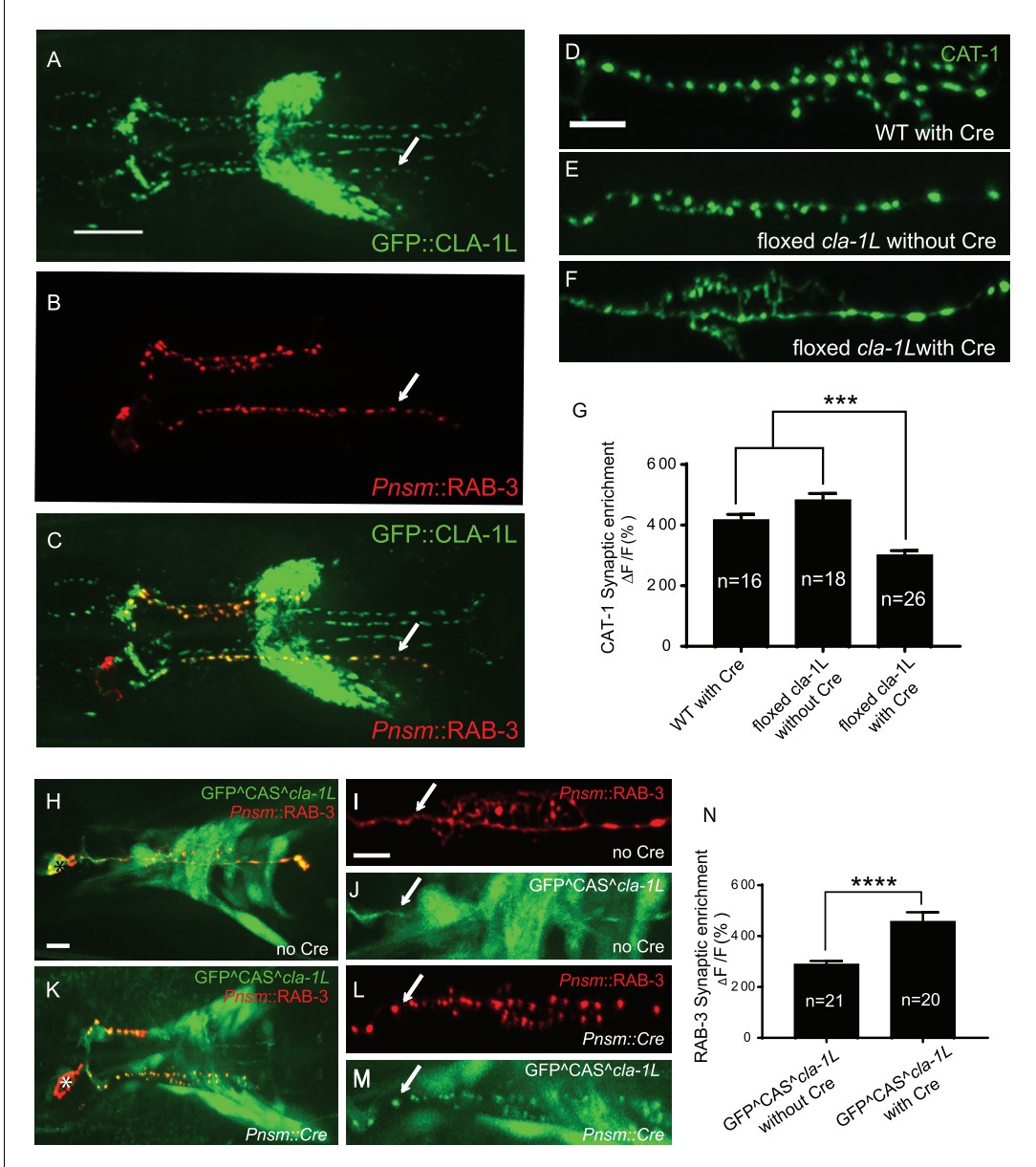

**Figure 2.** CLA-1 localizes to synapses and regulates synaptic vesicle clustering in a cell autonomous manner. (**A–C**) Endogenous expression of GFP:: CLA-1L (see Materials and methods) in the nerve ring of adult worms (**A**) along with NSM-specific expression of mCh::RAB-3 (**B**) CLA-1L colocalizes with RAB-3 in NSM (arrows) (**A,C**). Scale bar = 10 μm. (**D–F**) CAT-1::GFP distribution is normal in WT worms expressing Cre recombinase in NSM (**D**), and in floxed *cla-1L* worms without Cre (**E**), as expected. However, when Cre is expressed cell-specifically in NSM in the context of the floxed *cla-1L* allele, the synaptic vesicle pattern in NSM phenocopies that of *cla-1* loss-of-function mutants (**F**). Scale bar = 5 μm. (**G**) Synaptic enrichment (ΔF/F) of CAT-1::GFP in NSM for control animals ('WT with Cre' and 'floxed *cla-1L* without Cre'), and animals in which *cla-1L* was cell-specifically deleted in NSM ('floxed *cla-1L* with Cre'). n = number of animals. (**H–M**) Cytosolic GFP driven by the endogenous *cla-1L* promoter in place of CLA-1L (GFP^CAS^*cla-1L*; **H** and **J**) overlaps with RAB-3 expressed under the NSM promoter (*Pnsm*::RAB-3::mCh; **H** and **I**). RAB-3 shows defective vesicle clustering before Cre excision of the translation termination sequence (GFP^CAS^*cla-1*L without Cre; **I**; arrow). Upon cell-specific Cre expression in NSM (**K–M**), a functional, translational fusion of GFP:CLA-1L results (see Materials and methods and ***Figure 2—figure supplement 1***), rescuing the synaptic pattern in NSM (as determined by punctate distribution of both RAB-3 (**L**, arrow) and of GFP::CLA-1L (**M**, arrow)). Asterisk (**H** and **K**) corresponds to the location of the cell body of the NSM neurons. Scale bar = 5 μm. (**N**) Quantification of the synaptic enrichment (ΔF/F) of mCherry::RAB-3 in NSM for *cla-1l* null animals ('GFP^CAS^*cla-1*L without Cre') and animals expressing GFP::CLA-1L cell-specifically in NSM ('GFP^CAS^*cla-1*L with Cre'). n = number of animals.
DOI: https://doi.org/10.7554/eLife.29276.006

The following figure supplement is available for figure 2:

**Figure supplement 1.** Schematics of CRISPR strategies to examine subcellular localization and cell autonomy of CLA-1L.

*Figure 2 continued on next page*

*Figure 2 continued*

DOI: https://doi.org/10.7554/eLife.29276.007

CRISPR strain, CLA-1L colocalized with RAB-3 in NSM (*Figure 2A–C*), indicating that CLA1L localizes to synapses, at or near synaptic vesicle clusters.

## CLA-1 regulates synaptic vesicle clustering cell-autonomously

To determine whether CLA-1L regulates synaptic vesicle clustering cell-autonomously in NSM, we manipulated its expression in specific neurons using CRISPR-based strategies. Briefly, if CLA-1L acts cell-autonomously in NSM, cell-specific knockouts of CLA-1 should result in a cell-specific synaptic vesicle mutant phenotype, even in the context of all other cells expressing wild type CLA-1L. Conversely, in the context of all other cells lacking CLA-1L, cell-specific expression of wild type CLA-1L should result in cell-specific rescue of the synaptic vesicle phenotype.

To achieve cell-specific knockouts of CLA-1L, we created transgenic strains with loxP sites inserted within the introns flanking exon 3 and exon 13 of *cla-1L* (*Figure 1H* and *Figure 2—figure supplement 1B*). Insertion of loxP sites did not affect synaptic vesicle clustering in NSM, as predicted (*Figure 2E*). However, cell-specific expression of Cre in NSM, which leads to NSM-specific deletion of CLA-1L, resulted in the *cla-1L* mutant phenotype in NSM. Namely, we observed a diffuse distribution of synaptic vesicle proteins in NSM (*Figure 2F and G*). These findings indicate that CLA-1L is required in NSM for synaptic vesicle clustering and are consistent with it acting cell-autonomously in NSM.

To examine whether cell-specific expression of CLA-1L is sufficient to mediate synaptic vesicle clustering in *cla-1L* null mutant animals, we created a conditional *cla-1L*-expressing strain. We inserted a GFP followed by a transcriptional terminator before the start codon of *cla-1L* (*Figure 2—figure supplement 1C*). This construct drives GFP expression off the endogenous CLA-1L promoter, preventing the expression of the endogenous CLA-1L gene. In these animals, synaptic vesicle clustering was disrupted (*Figure 2I*, arrow) and GFP was observed throughout the nervous system, as predicated, and similar to transcriptional fusion transgenes previously examined (*Figure 2H* and *Figure 1—figure supplement 2A*). Cell-specific expression of Cre in NSM removes the transcriptional terminator and transforms it into an in-frame, functional translational fusion of the CLA-1L gene product (*Figure 2—figure supplement 1C*). In those animals, the resulting GFP::CLA-1L localized in a synaptic pattern in the NSM process (*Figure 2M*, arrow), colocalized with the synaptic vesicle marker RAB-3 (*Figure 2K–M*), and rescued the synaptic vesicle phenotype in NSM (*Figure 2L*, arrow, and N). Our findings indicate that CLA-1L is required cell-autonomously in the NSM neuron, where it is both necessary and sufficient to mediate synaptic vesicle clustering.

## CLA-1 isoforms regulate distinct aspects of synapse development at specific synapses

Given the broad expression pattern of *cla-1* in the nervous system (*Figure 1—figure supplement 2A–C*), we sought to determine whether CLA-1L functions to cluster synaptic vesicles in neurons other than NSM. We found that *cla-1(L)* mutants exhibited diffuse synaptic vesicle patterns in the AIY interneuron (*Figure 3A–D*) and the PVD mechanosensory neuron (*Figure 3—figure supplement 1A–C*), but not the GABAergic or cholinergic motor neurons that innervate body wall muscles (*Figure 3E–I and L*; *Figure 3—figure supplement 1D–F*), (although CLA-1L is expressed in at least a subset of these motor neurons; *Figure 1—figure supplement 2A*). These data demonstrate that CLA-1L is required for synaptic vesicle clustering at specific synapses in *C. elegans*, indicating that the molecular mechanisms for vesicle clustering may be cell (or synapse) specific.

*cla-1(S/M/L)* showed a similar phenotype to *cla-1(L)* in NSM (*Figure 1H*). Although *cla-1(S/M/L)* did not induce a diffuse synaptic vesicle phenotype in motor neurons either (*Figure 3H and I*), the number of synapses in these neurons was significantly reduced as compared to WT or to *cla-1(L)* mutants (*Figure 3J*). To more carefully quantify this effect, we examined the synaptic vesicle marker RAB-3 in a single cholinergic motor neuron, DA9 (*Figure 3K*). Consistent with our previous observations, we observed that *cla-1(S/M/L)* mutants have reduced numbers of RAB-3 puncta, suggesting a reduction in the number of synapses (*Figure 3L and M*). This is in contrast to mutants for the most

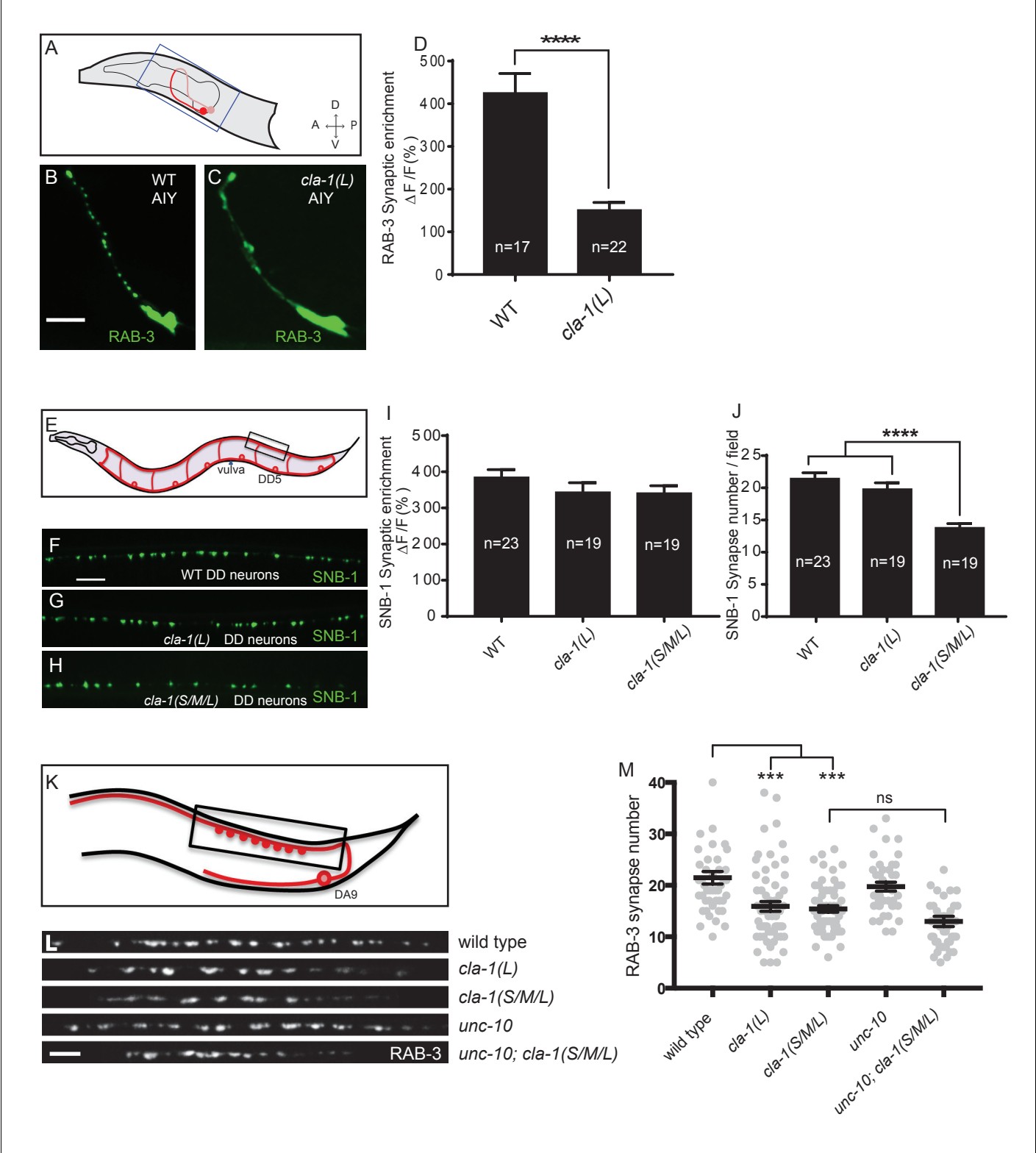

**Figure 3.** CLA-1 isoforms have discrete functions in several neuron types. (**A**) Schematic diagram of the bilaterally symmetric AIY interneuron (in red, within blue-boxed region) in the worm head. (**B–C**) RAB-3::GFP forms discrete presynaptic clusters in AIY of wild type animals (WT; **B**), but is diffuse in *cla-1(L)* mutants (**C**). Scale bar = 5 μm. (**D**) Synaptic enrichment (ΔF/F) of RAB-3::GFP in AIY for WT animals and *cla-1(L)* mutants. n = number of animals. (**E**) Schematic diagram of DD motor neurons. Synaptic vesicle clustering in DD neurons was assessed by examining the localization of SNB-1::GFP in the boxed area. (**F–H**). SNB-1::GFP forms discrete presynaptic clusters in DD axons of *cla-1(L)* or *cla-1(S/M/L)* mutants (**G–H**), similar to the wild type animals

*Figure 3 continued on next page*

Figure 3 continued

(WT; F). Scale bar = 10 μm. (I) Synaptic enrichment (ΔF/F) of SNB-1::GFP in the DD axons for WT animals and *cla-1(L)* or *cla-1(S/M/L)* mutants.
n = number of animals. (J) SNB-1::GFP puncta number in DD axons of *cla-1(S/M/L)* and *cla-1(L)* mutants, compared to WT animals. n = number of
animals. (K) Schematic of the DA9 cholinergic motor neuron. Synapses (boxed region) labeled by RAB-3::GFP were examined. (L) Straightened synaptic
domain (boxed region in K) showing the localization of RAB-3::GFP in WT animals and various mutants. Scale bar = 5 μm. (M) Synapse number was
reduced in *cla-1(S/M/L)* as well as *cla-1(L)* mutants compared to WT animals (although with greater variability in *cla-1(L)* mutants), but was not
significantly different between *cla-1(S/M/L)* single mutants and *cla-1(S/M/L);unc-10* double mutants.
DOI: https://doi.org/10.7554/eLife.29276.008
The following figure supplement is available for figure 3:

**Figure supplement 1.** Synaptic vesicle clustering assessed in various neuronal types.
DOI: https://doi.org/10.7554/eLife.29276.009

closely related synaptic gene *unc-10/RIM*, which did not show a decrease in synapse number or an
enhancement of *cla-1* (*Figure 3L and M*). We note that while DA9 motor neurons also display a
reduction in synapse number in *cla-1(L)* mutants, the expressivity of this phenotype was more vari-
able than that of the null allele (*Figure 3L and M*). Taken together, given the distinct synaptic phe-
notypes observed in different neurons, our results suggest that *cla-1* functions at specific synapses
to regulate different aspects of synaptic development.

## Distinct subsynaptic localization of different CLA-1 isoforms

To determine the subsynaptic localization of CLA-1, we tagged the CLA-1S cDNA with either N- or
C-terminal GFP and co-expressed it under a DA9 cell-specific promoter along with the synaptic vesi-
cle protein RAB-3 (*Figure 4A* and data not shown). N and C-terminal CLA-1S GFP fusion constructs
were indistinguishable and showed specific punctate localization at the ventral tip of the presynaptic
varicosity, where active zones (or their ultrastructural correlates, dense projections) are known to be
located from electron microscopy studies (*Stigloher et al., 2011*). Coexpression of CLA-1S with
ELKS-1 (*Figure 4B*) or with the calcium channel UNC-2 (*Figure 4C*) led to near complete colocaliza-
tion of CLA-1S with these active zone proteins, suggesting that CLA-1S specifically localizes to the
active zone.

   To determine the spatial relationship between CLA-1S and CLA-1L, we simultaneously labeled
CLA-1S (tagged with N-terminal mRuby3) and CLA-1L (endogenously tagged with N-terminal GFP)
and imaged their localization in DA9 neurons (*Figure 4D*; see Materials and methods for labeling
strategy of endogenous CLA-1L protein). As expected, both isoforms were enriched at the synapse.
However, unexpectedly, they differed regarding their subcellular localization within the synaptic
compartment. While the N-terminally tagged CLA-1S co-localized precisely with other active zone
proteins, N-terminally tagged CLA-1L displayed a more diffuse pattern of localization in the synaptic
region, away from the active zone. CLA-1L is a large (~9000 amino acid) protein, and its N-terminal
domain could lie far from its C-terminus. To examine this, we endogenously tagged the C-terminal
region of the cla-1 genomic locus, which would label the C-termini of all CLA-1 isoforms, including
CLA-1L. We observed that C-terminally tagged CLA-1 isoforms displayed a similar punctate pattern
and precise colocalization with the CLA-1S isoform (*Figure 4E*; note that additional puncta in the
second panel correspond to synapses in neurons other than DA9). Together our findings suggest
that CLA-1S and CLA-1L are anchored at the active zone via their C-terminus, and that the N-termi-
nus of CLA-1L may extend away from the active zone into other subcellular regions of the synaptic
bouton.

## *cla-1* mutants have ultrastructural defects in synaptic vesicle localization and dense projection morphology

To understand the ultrastructural organization underlying our light-level observations, we conducted
serial section electron microscopy (EM; *Figure 5A* and *Figure 5—figure supplement 1*). An average
of 130 wild type and 166 *cla-1(S/M/L)* mutant 40 nm sections were cut and reconstructed from three
worms from each genotype (encompassing 19 wild type and 12 mutant synapses). We found that
*cla-1* mutants had smaller terminal area size (*Figure 5B*), and fewer total synaptic vesicles (*Figure 5—
figure supplement 2A*) as compared to wild type animals. The vesicle density (vesicle number nor-
malized for terminal size) was indistinguishable between mutant and wild type animals (*Figure 5—*

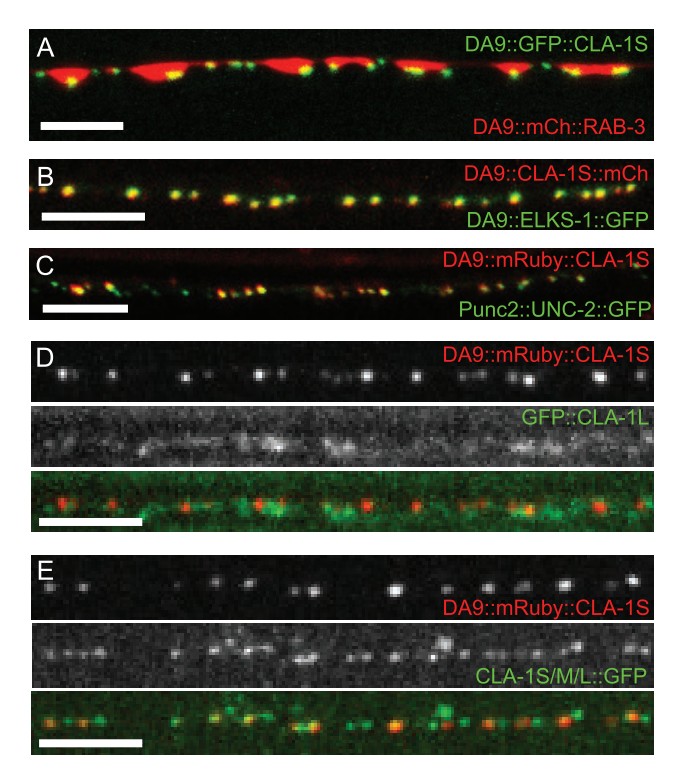

**Figure 4.** Subcellular localization of CLA-1 proteins. (**A–C**) CLA-1S localizes to the active zone. GFP::CLA-1S and mCherry::RAB-3 expressed in DA9 (**A**) show overlapping expression patterns, with CLA-1S fluorescence limited to a subregion of the RAB-3 domain. CLA-1S::mCherry or mRuby::CLA-1S expressed in DA9 colocalize well with ELKS-1::GFP (**B**) and the N-type calcium channel UNC-2::GFP (**C**), respectively. Scale bars = 5 μm. (**D**) mRuby3::CLA-1S expressed in DA9 along with endogenous expression of N-terminally tagged GFP::CLA-1L. Scale bar = 5 μm. (**E**) mRuby3::CLA-1S expressed in DA9 along with endogenous expression of C-terminally tagged CLA-1S/M/L::GFP. Scale bar = 5 μm.

DOI: https://doi.org/10.7554/eLife.29276.010

figure supplement 2B). The length of the dense projection was reduced in *cla-1* mutants (*Figure 5C*), suggesting a role for this protein in regulating the development of the dense projection. *cla-1* mutants also exhibited a reduction in the number of undocked synaptic vesicles contacting the dense projection (pseudocolored as pink vesicles in *Figure 5A*; *Figure 5D*), and a change in the distribution of docked vesicles (*Figure 5—figure supplement 2C*), including an increase in the number of docked vesicles within 100 nm of the dense projection (*Figure 5E*). Our findings suggest that CLA-1 is necessary for the development and clustering of synaptic vesicles at the dense projection, a region known to be crucial for vesicle release (*Stigloher et al., 2011*).

## *cla-1* mutants show defects in synaptic transmission

Defects in synaptic vesicle clustering or in the number of synaptic vesicle release sites frequently lead to changes in synaptic transmission (*Zhen and Jin, 1999*; *Hallam et al., 2002*). Defects in synaptic transmission can be quantitatively measured by resistance to the acetylcholinesterase inhibitor aldicarb, which potentiates the action of secreted acetylcholine (Ach) (*Mahoney et al., 2006*). Resistance to aldicarb is thus indicative of a reduction in secretion of ACh from cholinergic NMJs. Both *cla-1L(L)* and *cla-1(S/M/L)* mutants exhibited resistance to aldicarb, suggesting compromised synaptic transmission (*Figure 6—figure supplement 1A*). *cla-1* (*S/M/L*) animals were more resistant to aldicarb than *cla-1L(L)* (*Figure 6—figure supplement 1A*), suggesting that while the long isoform plays a role in synaptic transmission, the shorter isoforms and/or the C-terminus might execute additional functions that affect synaptic vesicle release.

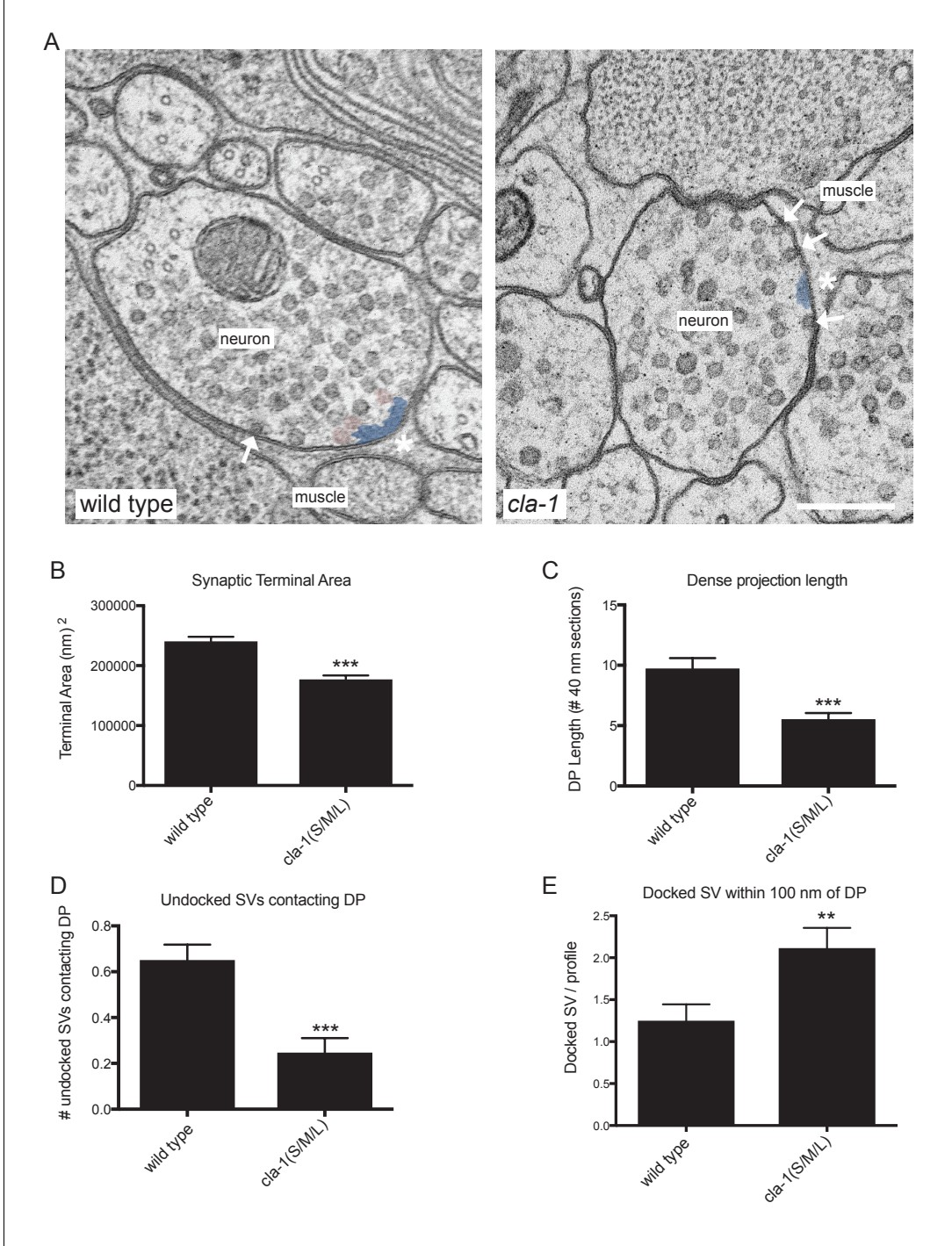

**Figure 5.** Ultrastructural analysis of *cla-1* mutants reveals changes in synaptic vesicle localization. (**A**) Representative micrographs of wild type and *cla-1 (S/M/L)* mutant synaptic profiles. Arrows indicate docked vesicles; asterisk indicates the dense projection (DP), which is also colored blue; undocked vesicles touching the dense projection are colored pink. Scale bar = 200 nm. (**B**) Synaptic terminal area (measured in nm²) is decreased in *cla-1(S/M/L)* mutants. (**C**) The length of the dense projection, measured in the number of 40 nm profiles in which it is observed, is decreased in *cla-1(S/M/L)* mutants. (**D**) The number of undocked synaptic vesicles directly in contact with the DP is reduced in *cla-1(S/M/L)* mutants. (**E**) The number of synaptic vesicles docked at the plasma membrane within 100 nm of the dense projection is increased in *cla-1(S/M/L)* mutants.

DOI: https://doi.org/10.7554/eLife.29276.011

The following figure supplements are available for figure 5:

**Figure supplement 1.** Ultrastructural analysis of *cla-1* mutants reveals changes in synaptic vesicle localization.
DOI: https://doi.org/10.7554/eLife.29276.012

*Figure 5 continued*

**Figure supplement 2.** Ultrastructural analysis of *cla-1* mutants reveals changes in synaptic vesicle localization.

DOI: https://doi.org/10.7554/eLife.29276.013

To determine more precisely how synaptic transmission was perturbed in the *cla-1* mutants, we recorded spontaneous and evoked responses in postsynaptic muscle cells using patch clamp electrophysiology. In *cla-1(S/M/L)*, but not in *cla-1(L)* mutants, the frequency of spontaneous postsynaptic currents ('minis') was reduced by 46% (*Figure 6A,C*). Since synapse number is also reduced in these mutants (*Figure 3M*), the reduction in mini frequency could be partially or wholly attributable to the reduction in synapse number. Mini amplitude was unchanged (*Figure 6D*), indicating that postsynaptic receptor function was not perturbed. While evoked response to a single presynaptic depolarization was unchanged in *cla-1* mutants (*Figure 6B,E*), subsequent release during a 20 Hz stimulation train was impaired (*Figure 6F* and *Figure 6—figure supplement 1B*), mirroring the aldicarb results. An increase in depression upon repeated stimulation indicates a defect in the number of vesicles that can be readily recruited by depolarization, and might be a functional consequence of the reduced number of vesicles contacting the dense projection (*Figure 5D*). Because *cla-1(L)* and *cla-1(S/M/L)* showed equally enhanced depression, we could not detect an additional role for the shorter CLA-1 isoforms. Our findings therefore suggest that the long isoform of CLA-1 might be solely responsible for synaptic vesicle recruitment to sustain release upon repetitive stimulation. Taken together our assays reveal functional consequences to the observed cell biological and ultrastructural phenotypes and suggest a specific role for CLA-1L in synaptic vesicle release in response to repeated depolarizations.

In light of the reduction in synapse number and mini frequency, as well as the increase in synaptic depression, we were surprised to see no defect in the response to a single evoked stimulus (*Figure 6E*). We hypothesized that this might reflect either a compensatory upregulation of vesicle release at the remaining synapses, or a redundancy between CLA-1 and another protein. Vesicle release is regulated by the related active zone protein UNC-10/RIM (*Wang et al., 1997*). To test the genetic relationship of UNC-10/RIM in the context of CLA-1 function and the physiological output of the synapse, we recorded from double mutants of *cla-1* and *unc-10/rim*. We found that *cla-1(S/M/L); unc-10/rim* double mutants showed reduced evoked release in response to a single stimulus when compared to *unc-10/rim* mutants alone (*Figure 6E*). Since we did not detect a change in the number of synapses between *unc-10;cla-1* double mutants as compared to *cla-1* mutants alone (*Figure 3M*), we interpret the enhanced defect in evoked release in the double mutants to be the result of functional requirement for both proteins at the active zone rather than a synthetic effect due to changes in synapse number.

## CLA-1 localization is dependent on syd-2/Liprin-α, syd-1 and unc-104/Kinesin-3

Active zone proteins not only colocalize with each other but also interact genetically in synapse development (*Patel et al., 2006*; *Van Vactor and Sigrist, 2017*). The scaffold molecule syd-2/Liprin-α and the rhoGAP syd-1/mSYD1A are among the first active zone proteins to arrive at the synapse (*Fouquet et al., 2009*), but the precise mechanisms through which these and other active zone proteins are trafficked to and localized at synapses is still largely unknown. To better understand the genetic relationship of CLA-1 to other active zone proteins and the molecular program that localizes CLA-1 to synapses, we examined CLA-1S localization in other active zone protein mutants as well as in mutants for the synaptic vesicle motor unc-104/Kinesin-3. We found that CLA-1S was greatly reduced, but not completely absent, in *unc-104*/Kinesin-3 mutants (*Figure 7A,C*), consistent with previous studies showing down-regulation of active zone proteins in *Drosophila kinesin-3* mutants (*Pack-Chung et al., 2007*; *Li et al., 2017*). Strikingly, CLA-1S was completely absent from the axon in mutants for the active zone scaffold protein SYD-2/Liprin-α, and greatly reduced in mutants for *syd-1*/mSYD1A (*Figure 7A,C*). *syd-2/Liprin-α* mutants have a profound defect in synaptic vesicle accumulation (*Zhen and Jin, 1999*; *Patel et al., 2006*; *Stigloher et al., 2011*), which can be assessed by the distribution of RAB-3 puncta in DA9 (*Figure 7—figure supplement 1A*). Consistent with CLA-1 being downstream of SYD-2 function in synaptic vesicle recruitment, double mutants of

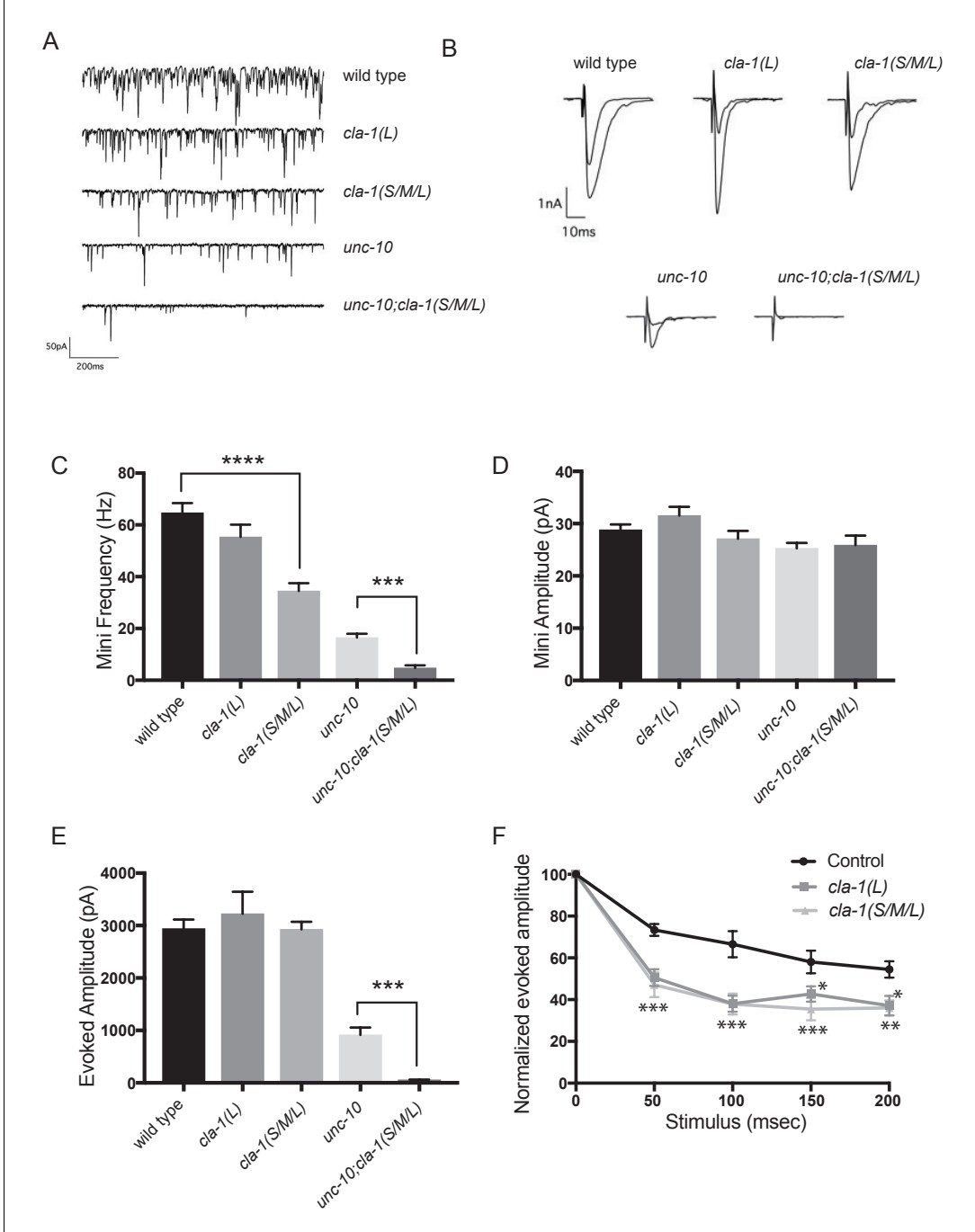

**Figure 6.** cla-1 mutant animals show defects in synaptic transmission. (**A**) Representative traces of endogenous post-synaptic current events. (**B**) Representative traces of evoked EPSCs, including the first and last recording from a five-stimulus train given at 20 Hz. (**C**) Frequency of endogenous miniature postsynaptic currents is reduced in *cla-1(S/M/L)* but not *cla-1(L)* mutants, compared to wild type. It is also further reduced in *cla-1(S/M/L);unc-10* double mutants compared to *unc-10* single mutants. (**D**) Amplitude of endogenous miniature postsynaptic currents is unchanged in *cla-1* and *unc-10* single and double mutants. (**E**) The amplitude of electrode-evoked responses to a single stimulus is unchanged in *cla-1* mutants compared to wild type, but is reduced in *cla-1(S/M/L);unc-10* double mutants when compared to *unc-10* single mutants. (**F**) Normalized amplitude of currents evoked by repeated electrode stimulation (interpulse interval = 50 msec) reveals increased depression in both *cla-1(S/M/L)* and *cla-1(L)* mutants.

DOI: https://doi.org/10.7554/eLife.29276.014

The following figure supplement is available for figure 6:

**Figure supplement 1.** cla-1 mutant animals show defects in synaptic transmission.
DOI: https://doi.org/10.7554/eLife.29276.015

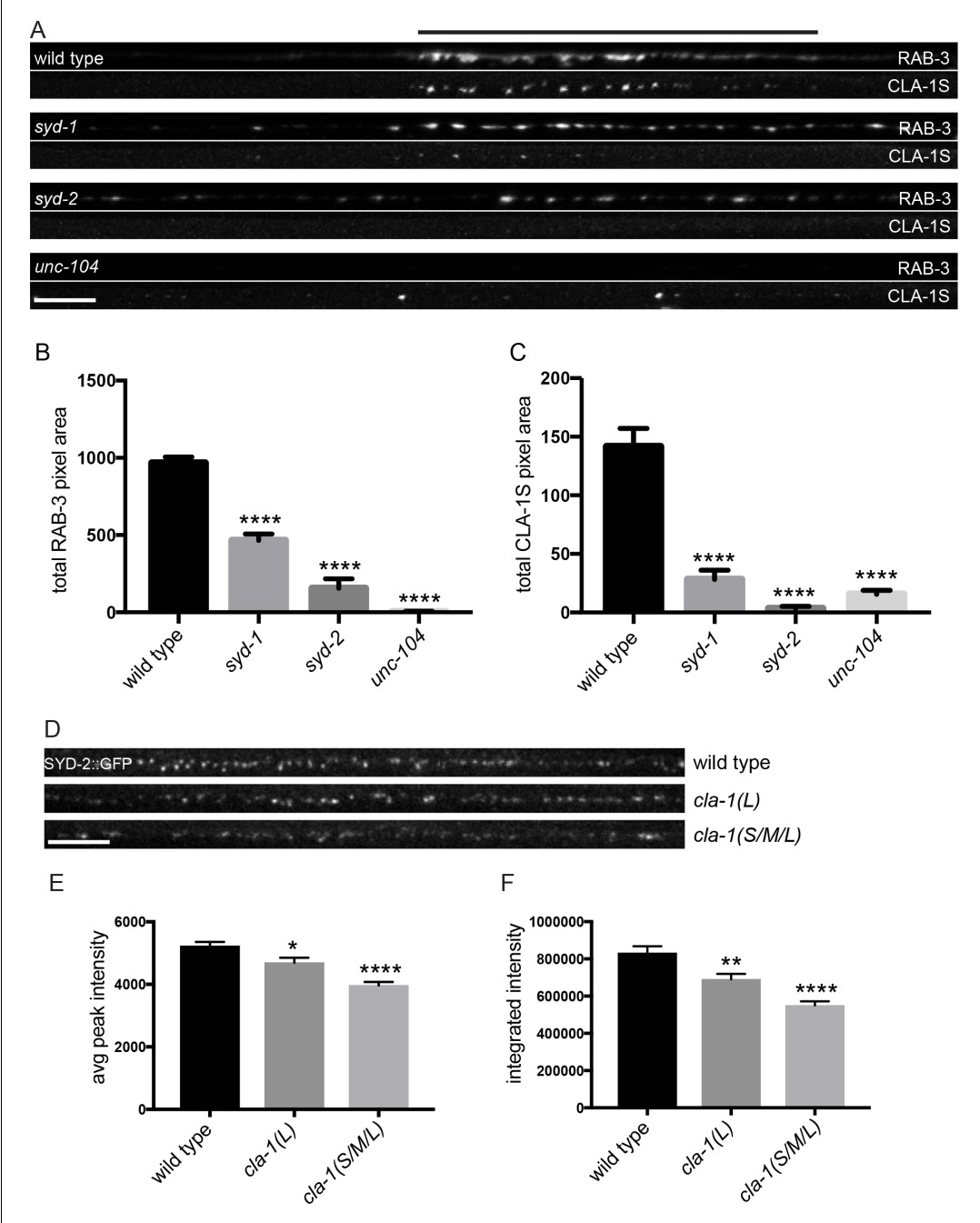

**Figure 7.** CLA-1S synaptic localization is regulated by UNC-104/Kinesin-3, SYD-2/liprin-α and SYD-1. (**A**) CLA-1S::GFP and mCherry::RAB-3 expression in the DA9 motor neuron of the indicated genotypes. *syd-1* and *syd-2/liprin-α* mutants exhibit smaller synaptic vesicle clusters that are distributed throughout the axon, and greatly reduced (in *syd-1*) or absent (in *syd-2*) CLA-1S puncta. No synaptic vesicles are detected in *unc-104* mutant axons, while the number of CLA-1S puncta is greatly diminished. Scale bar = 5 μm. Line above images indicates wild type synaptic domain. (**B**) Average total pixel area of mCherry::RAB-3 for wild type animals and various mutants. (**C**) Average total pixel area of CLA-1S::GFP for wild type animals and various mutants. (**D**) Endogenous SYD-2::GFP expression in motor neurons of the posterior dorsal nerve cord is reduced in *cla-1* mutants. Scale bar = 5 μm. (**E**) Average SYD-2::GFP puncta intensity is reduced in *cla-1* mutants. (**F**) SYD-2::GFP total (integrated) intensity normalized to length is reduced in *cla-1* mutants.

DOI: https://doi.org/10.7554/eLife.29276.016

The following figure supplement is available for figure 7:

**Figure supplement 1.** CLA-1L synaptic localization is regulated by UNC-104/Kinesin-3, SYD-2/liprin-α and SYD-1.
DOI: https://doi.org/10.7554/eLife.29276.017

*cla-1* and *syd-2* did not show a detectable enhanced synaptic phenotype, or additional synthetic phenotypes (*Figure 7—figure supplement 1A*). CLA-1S localization was also tested in several other synaptic mutants, including *elks-1*, *unc-10*/RIM and *rimb-1*/RIM-BP (and triple mutants for all three of these genes), but we could not detect a requirement for these genes in proper localization of CLA-1S (data not shown).

We also examined whether *unc-104/Kinesin-3*, *syd-1* and *syd-2/Liprin-α* mutants regulate the localization of endogenous CLA-1L. Since our CRISPR-tagged strain labels CLA-1L in many neurons, we were not able to examine CLA-1L distribution with single-cell resolution and assayed instead localization of these active zone proteins to the synapse-rich regions of the nerve ring. Consistent with our cell-specific analyses using CLA-1S, all three mutants resulted in reduced CLA-1L intensity at the nerve ring (*Figure 7—figure supplement 1B–E*). Taken together, these results show that CLA-1 localization at synapses is dependent on SYD-2/Liprin-α and SYD-1, but is independent of other active zone genes such as ELKS-1 and UNC-10/RIM.

To determine whether loss of CLA-1 may itself affect active zone composition, we examined the synaptic distribution of endogenously tagged SYD-2/Liprin-α::GFP (*Figure 7D*). Endogenous SYD-2 puncta in the dorsal nerve cord were dimmer in *cla-1* mutants (*Figure 7E*), and the overall fluorescence intensity was reduced (*Figure 7F*). To gain cellular specificity, we examined the localization of GFP-tagged SYD-2/Liprin-α expressed in NSM in *cla-1* mutants (*Figure 7—figure supplement 1F–I*). We found that in *cla-1(S/M/L)* but not *cla-1(L)* mutants, SYD-2::GFP localization was more diffuse (*Figure 7—figure supplement 1F–I*). Together these data demonstrate that loss of CLA-1 affects the recruitment or maintenance of SYD-2/Liprin-α at active zones.

## Discussion

Here, we report the discovery and characterization of a novel active zone protein in *C. elegans*, Clarinet (CLA-1) required for proper synapse number and function. The different isoforms of clarinet serve both to cluster vesicles at synapses and to recruit and release them at the active zone.

### CLA-1 isoforms have distinct roles in synapse development and function

In this study, we used two deletion alleles to interrogate the function of *cla-1*. The *cla-1(L)* allele specifically deletes the start of the long isoform, but does not affect the short and medium isoforms. The *cla-1(S/M/L)* allele deletes the PDZ and C2 domain-containing C-terminus shared by all three isoforms. By comparing phenotypes between these two alleles, we were able to assign distinct roles to the N-terminus of the long isoform, versus the common C-terminus, or the short/medium isoforms.

Spontaneous synaptic vesicle release as well as inhibitory motor neuron synapse number were impaired in *cla-1(S/M/L)*, but not in *cla-1(L)*, suggesting that either the common C-terminus or only the CLA-1S/M isoforms are involved in these processes. However, the *cla-1(L)*-specific mutant exhibited synaptic vesicle clustering defects in many sensory neurons, and synaptic transmission defects upon repetitive stimulation as well as increased aldicarb resistance. These data suggest that the long CLA-1 isoform is specifically required for synaptic vesicle clustering and functions during periods of sustained release. Importantly, our findings indicate that distinct CLA-1 isoforms might play specific roles to promote synaptic development and function.

### Subsynaptic localization of CLA-1 isoforms and their role in vesicle clustering and release

CLA-1S (whether it is N- or C-terminally tagged) colocalizes with active zone proteins. CLA-1L and CLA1S share the same C-terminal PDZ and C2 domains with sequence homology to vertebrate active zone proteins Piccolo and RIM. An endogenous C-terminal tag of all CLA-1 isoforms colocalizes with CLA-1S. These findings suggest that all CLA-1 isoforms may be anchored at the active zone by their C-terminus.

The N-terminally tagged CLA-1L still localized to synaptic areas but was not confined to the synaptic subregions occupied by CLA-1S and known active zone proteins. CLA-1L is a large,~9000 amino acid protein that, if anchored to the active zone area via its C-terminus, could possibly extend away to regions occupied by undocked synaptic vesicles. Because the sub-synaptic localization of

the N-terminally tagged CLA-1L differed from that of the C-terminally tagged CLA-1 isoforms, our findings are consistent with a model in which the N- and C-termini of CLA-1L occupy distinct sub-synaptic areas (*Figure 8*). This model is analogous to the orientation of *Drosophila* BRP at the fly neuromuscular junction (*Fouquet et al., 2009*) and consistent with models of Piccolo as a protein oriented in a polarized manner and extending ~100 nm from the plasma membrane (*Dani et al., 2010*). Since Clarinet is almost twice the size of Piccolo and exhibits more unstructured regions, it could potentially extend even farther.

Our model is also consistent with the genetic and electrophysiological roles we identify for the long CLA-1L isoform in clustering vesicles at sensory synapses and possibly recruiting vesicles for release upon repeated stimulation. *C. elegans* Liprin-α, *Drosophila* BRP and Fife and mammalian Piccolo and Bassoon have all been implicated in clustering synaptic vesicles at the active zone (*Stigloher et al., 2011*; *Kittelmann et al., 2013*; *Hallermann et al., 2010*; *Mukherjee et al., 2010*; *Bruckner et al., 2012*; *Bruckner et al., 2017*). Deleting just the last 17 amino acids of BRP leads to the complete loss of synaptic vesicles adjacent to the T bar, as well as increased synaptic depression (*Hallermann et al., 2010*), suggesting that the inability of BRP to tether synaptic vesicles to the T bar contributes directly to the sustained release defect upon repeated stimulation. Our model would indicate that while N-terminal protein sequence between active zone proteins Clarinet, BRP, Fife and Piccolo/Bassoon varies, they share analogous molecular architecture required to link the synaptic vesicle pool with the active zone to actuate their function at presynaptic sites.

The smaller size of the dense projection in *cla-1* mutants indicates that this protein is either a component of this presynaptic specialization, or is required for its development. The *C. elegans* dense projection is thought to organize synaptic vesicles and their release machinery, much like the *Drosophila* T bar and the ribbon structure in the mammalian visual system. We observe a co-dependency between CLA-1 and the active zone protein SYD-2 in their recruitment to the synapse, consistent with a requirement of these proteins for the assembly of the dense projection.

## Interaction between CLA-1 and UNC-10/RIM in synaptic vesicle release

Despite the presence of morphological and structural abnormalities, *cla-1* mutants exhibit normal responses to single evoked stimuli. This may be due either to compensatory upregulation of vesicle release at the remaining synapses, or to redundancy with another active zone protein. Mutants for active zone proteins implicated in synaptic vesicle release, such as RIM/UNC-10 and UNC-13, exhibit

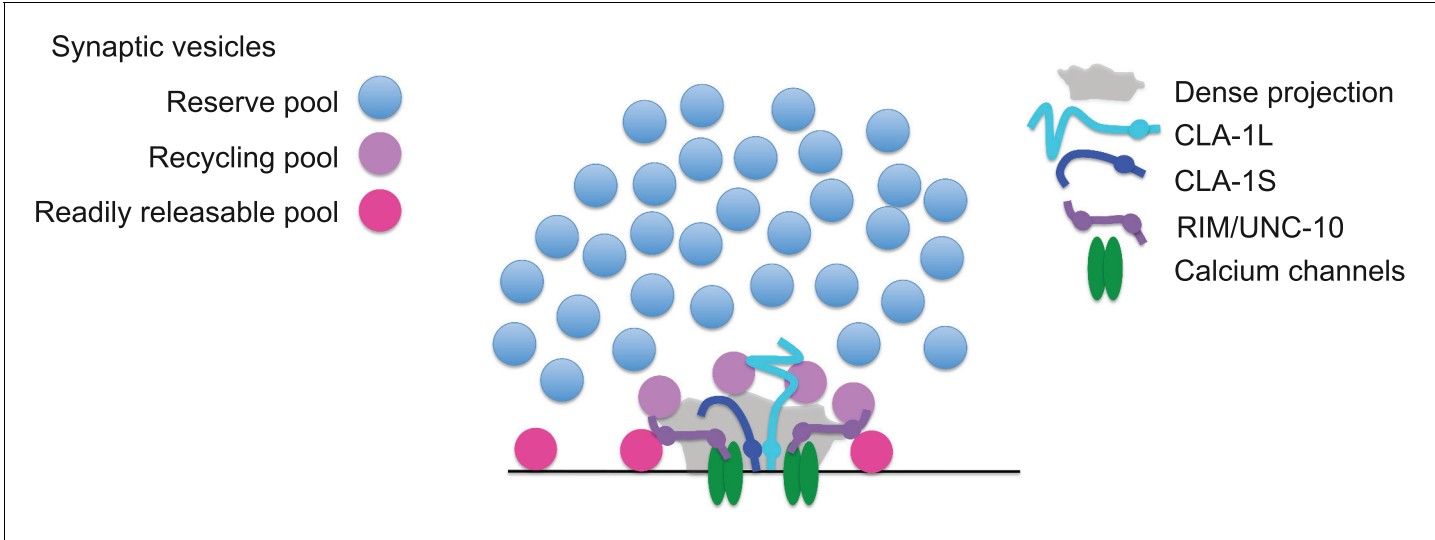

**Figure 8.** Model for how CLA-1 isoforms may mediate distinct synaptic vesicle interactions. A model for how CLA-1 S (dark blue) and L (turquoise) isoforms and RIM/UNC-10 (purple) may be organized at the active zone and interact with synaptic vesicles. CLA-1S localizes to the dense projection. The C-terminus of CLA-1L is also anchored at the dense projection, but its N-terminus extends out into the synaptic bouton. Interaction of CLA-1L with synaptic vesicles may facilitate their clustering at the dense projection and release upon repeated stimulation.
DOI: https://doi.org/10.7554/eLife.29276.018

a reduction in the number of docked synaptic vesicles at the active zone (*Stigloher et al., 2011*; *Weimer et al., 2006*; *Gracheva et al., 2008*; *Kaeser et al., 2011*; *Han et al., 2011*; *Wang et al., 2016*; *Acuna et al., 2016*). RIM/UNC-10 in particular localizes within 100 nm of the dense projection and is responsible for vesicle docking in this region (*Weimer et al., 2006*; *Gracheva et al., 2008*), precisely where *cla-1* mutants exhibit an increase in docked vesicles. This increase in morphologically docked vesicles might be the structural correlates of a compensatory upregulation of primed vesicles. Consistent with this model, in the absence of UNC-10/RIM, loss of CLA-1 further reduces evoked responses after a single stimulus, suggesting that UNC-10/RIM could be responsible for a compensatory response in *cla-1* mutants.

Alternatively, it is also possible that docked synaptic vesicles accumulate in *cla-1* mutants due to a *cla-1*-dependent release defect. BRP, Fife, Rim and Bassoon have all been shown to play a role in calcium channel localization (*Bruckner et al., 2017*; *Kaeser et al., 2011*; *Han et al., 2011*; *Kittel et al., 2006*; *Graf et al., 2012*; *Frank et al., 2010*), and both *Drosophila* Fife and mammalian rim mutant phenotypes are consistent with an impairment in the coupling of synaptic vesicles to calcium channels (*Bruckner et al., 2017*; *Kaeser et al., 2011*; *Han et al., 2011*). Regardless of the cause of the additive phenotype in the *unc-10;cla-1* double mutants, our findings indicate a genetic, and functionally significant interaction between CLA-1 and a protein known to function in synaptic vesicle release, UNC-10/RIM. Together these data underscore the functional consequences of loss of CLA-1 at the synapse.

## Role of CLA-1 in synaptic vesicle clustering

How synaptic vesicles are clustered at synapses remains poorly understood. Initial studies suggested that synapsin tethers synaptic vesicles to the actin cytoskeleton (*Bähler et al., 1990*), but more recent evidence calls that model into question (*Pechstein and Shupliakov, 2010*; *Shupliakov et al., 2011*) and suggests that other as yet unidentified proteins may be involved in synaptic vesicle clustering (*Siksou et al., 2007*; *Fernández-Busnadiego et al., 2010*; *Stavoe and Colón-Ramos, 2012*; *Stavoe et al., 2012*). Mammalian Piccolo has been shown to play a role in recruiting synaptic vesicles from the reserve pool through interactions with synapsin (*Leal-Ortiz et al., 2008*; *Waites et al., 2011*), and to maintain synaptic vesicle clustering at the active zone (*Mukherjee et al., 2010*), while SYD-2 has been shown to cluster vesicles by influencing transport (*Edwards et al., 2015*). Tomosyn has also been shown to regulate synaptic vesicle distribution between the reserve and recycling pools, possibly through interactions with synapsin (*Cazares et al., 2016*).

CLA-1L, which extends away from the active zone, may be an important link in understanding how synaptic vesicles are clustered and recruited. Our analyses of synaptic vesicle clustering at various synapses by confocal microscopy indicated that CLA-1L was required to cluster synaptic vesicles at synapses in several different classes of neurons, although not in excitatory or inhibitory motor neurons. Our ultrastructural analysis and functional assays revealed that at motor neuron synapses, CLA-1 is involved in tethering vesicles to the dense projection and CLA-1L itself is implicated in recruiting synaptic vesicles for release upon repeated stimulations. Our findings suggest that although CLA-1L might not display a change in synaptic vesicle clustering by fluorescence measurements in motor neurons, it could still play a role in synaptic vesicle recruitment to the active zone at these synapses. We speculate that CLA-1L may retain the recycling pool of vesicles (i.e. vesicles recruited upon multiple stimulations) at the dense projection (*Figure 8*). In certain neurons (including NSM, AIY and PVD), this may lead in turn to the retention of the reserve pool of vesicles within the presynaptic bouton.

## CLA-1 isoforms encode a novel set of proteins with conserved functional roles at the active zone

Of all the isoforms, CLA-1L is the most enigmatic due to its large size and structure. Almost half of CLA-1L consists of a repetitive region, which is predicted to be disordered and has no sequence homology to vertebrate proteins. The structure, function, regulation and evolution of the repetitive region pose interesting questions. The distribution of this protein within the synaptic bouton and its function in synaptic vesicle release suggest a novel mechanism for clustering synaptic vesicles, with shared functional homology to vertebrate and *Drosophila* active zone proteins. The mechanisms

uncovered in this study might therefore demonstrate how divergent strategies can be utilized for conserved purposes in organizing the development and function of synapses.

## Materials and methods

### Strains and genetics

Worms were raised on NGM plates at 20°C using OP50 *Escherichia coli* as a food source. N2 Bristol was used as the wild type reference strain. Hawaii CB4856 strain was used for SNP mapping. The following mutant strains were obtained through the Caenorhabditis Genetics Center: *cla-1(ok560)IV*, *cla-1(gk352)IV, cla-1(ok937)IV, cla-1(ok2285)IV, unc-104(e1265)II, syd-2(ok217)X, syd-2(ju37)X, syd-1 (ju82)II*, *unc-10(md1117)X* and zxIs6 [unc-17p::ChR2(H134R)::YFP + lin-15(+)] V. nuIs168 [Pmyo-2::gfp + Punc-129::Venus::rab-3] was provided by Jihong Bai (Fred Hutchinson Cancer Research Center, Seattle, Washington). juIs137 [Pflp-13::snb-1::gfp] was provided by Yishi Jin (UCSD, San Diego, CA). kyIs445 [Pdes-2::mCherry::rab-3 + Pdes-2:sad-1::gfp] was provided by Cori Bargmann (Rockefeller University, New York, NY). Other strains used in the study are as follows: olaIs1 [Ptph-1::mCherry + Ptph-1::cat-1::gfp], olaEx3222 [Ptph-1::mCherry::rab-3]; cla-1(ola311)IV [GFP::CLA-1L], olaEx3309 [Ptph-1::mCherry + Ptph-1::cat-1::gfp; Ptph-1::cre]; cla-1(ola324)IV [floxed cla-1L], olaEx3289 [Ptph-1::mCherry::rab-3 + Ptph-1::cre]; cla-1(ola321)IV [GFPCAScla-1L], olaEx2897 [Pcla-1L::gfp + Prab-3::mCherry], olaEx2898 [Pcla-1M::gfp + Prab-3::mCherry], olaEx2924 [Pcla-1S::gfp + Prab-3::mCherry], olaEx1106 [Ptph-1:: mCherry::rab-3 + Ptph-1:syd-2::gfp], wyIs45 [Pttx-3::rab3::gfp], wyIs85 [Pitr-1:: GFP::RAB-3], wyIs574 [Pmig-13::CLA1S::GFP], wyIs226 [Pmig-13::mCherry::RAB-3], wyEx8596 [Pmig-13::mRuby3::CLA-1S], wyEx6368 [Pmig-13::CLA-1S::mCherry + Pmig-13::GFP::ELKS-1], wyEx9404 [Pmig13::FLPase + Pmig13::mRuby3::cla-1];cla-1(wy1186)IV [C-terminal FRT-stop-FRT GFP], *syd-2 (wy1074)* [endogenous N-term GFP].

### Molecular biology and transgenic lines

Expression clones were made in the pSM vector (*Shen and Bargmann, 2003*). The plasmids and transgenic strains (0.5–50 ng/µl) were generated using standard techniques and coinjected with markers Punc122::GFP (15–30 ng/µl), Punc122::dsRed (15–30 ng/µl), Podr-1::RFP (100 ng/µl) or Podr-1::GFP (100 ng/µl).

### Screen and SNP mapping coupled with WGS

Worms expressing CAT-1::GFP and cytosolic mCherry in NSM neuron (olaIs1) were mutagenized with ethyl methanesulfonate (EMS) as described previously (*Brenner, 1974*). The screen was performed as previously described (*Nelson and Colón-Ramos, 2013*; *Jang et al., 2016*). CAT-1::GFP was diffusely distributed throughout neurites in six mutants, including *cla-1(ola104)*. The *ola104* allele was mapped to a 2.1Mbp region on chromosome IV using SNP mapping coupled with whole-genome sequencing (WGS) (*Davis et al., 2005*; *Doitsidou et al., 2010*). WGS identified the genetic lesion in ola104 as a missense mutation in cla-1. ola104/cla-1(ok560) trans-heterozygotes were examined for complementation.

### Phylogenetic tree creation

We generated a phylogenic tree to determine how related the CLA-1 PDZ domain was to the other family members (*Figure 1I*). The PDZ domains of Piccolo/Fife-related proteins were identified by SMART (*Schultz et al., 1998*; *Letunic et al., 2012*). T-Coffee (M-Coffee) was used for multi-alignment of the sequences (*Notredame, 2010*). A rooted phylogenetic tree was determined from aligned sequences by neighbor joining with 100 bootstrap replicates using APE (*Paradis et al., 2004*). PDZ domains of Dishevelled family proteins were used as an outgroup. A circle tree was built using ggtree (*Yu et al., 2016*).

### RT -PCR

RNA from wild type, *cla-1(S/M/L)* and *cla-1(L)* worms was prepared using Trizol (Sigma Aldrich, St. Louis, MO). A cDNA library was created by reverse transcription using oligo dTs. PCR amplification was conducted using primers against the C-terminal PDZ domain of *cla-1*, as well as against the housekeeping gene *tba-1*.

## Fluorescence microscopy and confocal imaging

Images of fluorescently tagged fusion proteins were captured at room temperature in live *C. elegans*. Mid-L4 through young adult stage hermaphrodite animals were anesthetized using 10 mM levamisole (Sigma-Aldrich) or 50 mM muscimol (Abcam) in M9 buffer, mounted on 2–5% agar pads and imaged as follows: Images in *Figures 1*, *2* and *3B*-Hwere taken using a 60x CFI Plan Apochromat VC, NA 1.4, oil objective (Nikon) on an UltraView VoX spinning-disc confocal microscope (PerkinElmer). Images in *Figures 4A–C and* and *7A* were taken using a Zeiss LSM710 confocal microscope (Carl Zeiss) with a Plan-Apochromat 63x/1.4 NA objective. Images in *Figures 3L*, *4D–E and* and *7D* were taken with a Zeiss Axio Observer Z1 microscope equipped with a Plan-Apochromat 63 × 1.4 objective and a Yokagawa spinning-disk unit. Maximum-intensity projections were generated using ImageJ (NIH) or ZEN 2009 software and used for all the confocal images. Quantification was performed on maximal projections of raw data.

## Quantification of synaptic vesicle clustering and synapse number phenotypes

Quantification of synaptic vesicle clustering in *Figures 1–3* and active zone protein clustering in *Figure 7—figure supplement 1* was based on a previous protocol (*Jang et al., 2016*). Briefly, fluorescence values for individual neurites (ventral neurite for the NSM and PVD neurons, Zone3 for the AIY neuron, and dorsal neurite for DD GABAergic or cholinergic motor neurons) were obtained through segmented line scans using ImageJ. A sliding window of 2 µm was used to identify all the local fluorescence peak values and trough values for an individual neuron. Synaptic enrichment was then calculated as % ΔF/F as previously described (*Dittman and Kaplan, 2006*; *Bai et al., 2010*). To measure penetrance, animals were scored as displaying either 'punctate' or 'diffuse' phenotypes for synaptic vesicles proteins. Percentage of animals displaying diffuse distribution of synaptic vesicle proteins was calculated for each genotype. For each experiment, at least 30 animals were scored for each genotype and at least five independent experiments were performed. The number of synaptic vesicle puncta in DD GABAergic motor neurons was counted by ImageJ with the same settings for all images including threshold, size and circularity. DA9 synapse number in *Figure 3* and SYD-2::GFP puncta fluorescence in *Figure 7* was quantified using a Matlab (Mathworks, Natick, MA) script that counted and measured peaks above threshold from plot profiles of segmented line scans generated in ImageJ. To quantify synaptic fluorescence of CLA-1S or RAB-3 in *Figure 7*, total integrated intensity of the line scans was analyzed using an ImageJ plugin.

## Generation of *cla-1(S/M/L)*

To create *cla-1(wy1048)* we chose sgRNAs ~13 kb apart designed to delete most of the M and almost all of the S isoform, including the shared PDZ and C2 domains. sgRNAs were injected at 30 ng/µl along with Cas9 plasmid at 50 ng/µl and F2 worms were screened by PCR. The resulting deletion is flanked by the following sequences: 5' CCACAACAATCATTCCACCC, 3' AGGTGTCGGCACACGTCATC.

## N-terminal endogenous labeling of CLA-1L

To endogenously tag CLA-1L at the N-terminus, a CRISPR protocol (*Dickinson et al., 2015*) was used to create cla-1(ola300[gfp:: SEC::cla-1L]), in which gfp::SEC (Self-Excising Cassette) was inserted before the start codon of cla-1L (*Figure 2—figure supplement 1A*). SEC consists of a hygromycin resistance gene (hygR), a visible marker [sqt-1(d)]) and an inducible Cre recombinase (*Figure 2—figure supplement 1A*). SEC is flanked by LoxP sites, and heat shock induced Cre expression removed the SEC, leaving GFP fused to CLA-1L in *cla-1(ola311[gfp::cla-1L])* (*Figure 2—figure supplement 1A*).

## Cell autonomy of CLA-1L

Two methods were used to demonstrate cell autonomy of CLA-1L. In the first method, a CRISPR protocol (*Paix et al., 2014*; *Arribere et al., 2014*) was used to create cla-1 (ola324), in which two loxP sites were inserted into two introns of cla-1L (*Figure 1G* and *Figure 2—figure supplement 1B*). We used three criteria to ensure that our insertion sites efficiently and specifically target CLA-1L. First, we avoided inserting loxP sites into small introns to prevent any effects on splicing. Second,

to ensure that CLA-1M is unaffected after Cre-loxP recombination, the second loxP site was positioned about 4 kb away from the start codon of cla-1M. Third, the sequence flanked by loxP sites is about 16 kb and is close to the start codon of cla-1L. Thus removal of the sequence should result in a CLA-1L null mutation. Cell-specific removal of CLA-1L in NSM was achieved with a plasmid driving the expression of cre cDNA under the NSM-specific *tph-1* promoter fragment as described previously (*Jang et al., 2016*; *Nelson and Colón-Ramos, 2013*).

In the second method, we modified a CRISPR protocol (*Dickinson et al., 2015*) to create cla-1 (ola321[gfp:: CAS::cla-1L]), in which CAS consists of a hygromycin resistance gene (hygR) and a visible marker [sqt-1(d)]) (*Figure 2—figure supplement 1C*). Since CAS contains a transcriptional terminator, this strain is a *cla-1*L null allele. Since CAS is flanked by loxP sites, Cre-loxp recombination generates functional GFP fused to CLA-1L. Cell-specific rescue in NSM was achieved with a plasmid driving the expression of cre cDNA under the NSM-specific *tph-1* promoter fragment. Detailed subcloning information will be provided upon request.

## C-terminal endogenous tagging of CLA-1 isoforms

A cell-specific CRISPR protocol (*Schwartz and Jorgensen, 2016*) was used to insert a let-858 3'UTR flanked by FRT sites followed by GFP at the conserved C-terminus of cla-1. Upon crossing to a strain containing cell-specific FLPase, the endogenous stop site and exogenous 3'UTR are excised, leaving the C-terminal GFP inserted in front of the endogenous 3'UTR. To achieve DA9-specific expression of CLA-1::GFP we used a FLPase driven by the Pmig-13 promoter, which has previously proven to be specific to DA9 within the posterior dorsal cord. However, Pmig-13 seems to express at very low levels in other neurons in this region (enough to generate excision at the FRT sites), as evidenced by the fact that we see CLA-1::GFP puncta outside DA9 driven exogenously expressed CLA-1 (*Figure 4E*).

## Aldicarb assays

Animals were assayed for acute exposure to aldicarb (*Mahoney et al., 2006*). Aldicarb (ULTRA scientific) was prepared as a stock solution of 200 mM stock in 50% ethanol. Aldicarb sensitivity was measured by transferring 25 animals to plates containing 1 mM aldicarb and then assaying the time course of paralysis. Animals were considered paralyzed once they no longer moved even when prodded with a platinum wire three times on the head and tail. The ratio of animals moving to the total number of animals on the plate was calculated for each time point. All strains used for this assay also contained zxIs6 in the background for consistency with electrophysiology assays. All assays were performed blinded to genotype.

## Electrophysiology

Electrophysiological recordings were obtained from the *C. elegans* neuromuscular junctions of immobilized and dissected adult worms as previously described (*Richmond, 2009*). Ventral body wall muscle recordings were acquired in whole-cell voltage-clamp mode (holding potential, −60 mV) using an EPC-10 amplifier, digitized at 1 kHz. Evoked responses were obtained using a 2 ms voltage pulse applied to a stimulating electrode positioned on the ventral nerve cord anterior to the recording site. For multiple stimulations, a five pulse train was delivered at 20 Hz. The 5 mM $Ca^{2+}$ extracellular solution consisted of 150 mM NaCl, 5 mM KCl, 5 mM $CaCl_2$, 4 mM $MgCl_2$, 10 mM glucose, 5 mM sucrose, and 15 mM HEPES (pH 7.3,~340 mOsm). The patch pipette was filled with 120 mM KCl, 20 mM KOH, 4 mM $MgCl_2$, 5 mM (N-tris[Hydroxymethyl] methyl-2-aminoethane-sulfonic acid), 0.25 mM $CaCl_2$, 4 mM $Na^2ATP$, 36 mM sucrose, and 5 mM EGTA (pH 7.2,~315 mOsm). Data were obtained using Pulse software (HEKA. Subsequent analysis and graphing was performed using mini analysis (Synaptosoft), Igor Pro and Prism (GraphPad).

## Electron microscopy

Worms underwent high-pressure freeze (HPF) fixation as described previously (*Weimer, 2006*). Young adult hermaphrodites were placed in specimen chambers filled with *Escherichia coli* and frozen at −180°C and high pressure (Leica SPF HPM 100). Samples then underwent freeze substitution (Reichert AFS, Leica, Oberkochen, Germany). Samples were held at −90°C for 107 hr with 0.1% tannic acid and 2% $OsO_4$ in anhydrous acetone. The temperature was then increased at 5 °C/h to

−20℃, and kept at −20℃ for 14 hr, and increased by 10 °C/h to 20℃. After fixation, samples were infiltrated with 50% Epon/acetone for 4 hr, 90% Epon/acetone for 18 hr, and 100% Epon for 5 hr. Finally, samples were embedded in Epon and incubated for 48 hr at 65℃. All specimens were prepared in the same fixation and subsequently blinded for genotype. Ultra thin (40 nm) serial sections were cut using an Ultracut 6 (Leica) and collected on formvar-covered, carbon-coated copper grids (EMS, FCF2010-Cu). Post-staining was performed using 2.5% aqueous uranyl acetate for 4 min, followed by Reynolds lead citrate for 2 min. Images were obtained on a Jeol JEM-1220 (Tokyo, Japan) transmission electron microscope operating at 80 kV. Micrographs were collected using a Gatan digital camera (Pleasanton, CA) at a magnification of 100 k. Images were quantified blinded to genotype using NIH ImageJ software and macros provided by the Jorgensen lab. Data were analyzed using MATLAB scripts written by the Jorgensen lab and Ricardo Fleury.

Images of the dorsal cord were taken for three animals from each strain. Cholinergic synapses were identified by morphology (*White et al., 1986*). A synapse was defined as a set of serial sections containing a dense projection and two flanking sections without dense projections from either side. Synaptic vesicles were identified as spherical, light gray structures with an average diameter of ~30 nm. To control for inherent variability in the size of synaptic terminals, we measured the density of synaptic vesicles in the terminal by dividing the number of synaptic vesicles by the area of the terminal in micrometers. Terminal area was defined as the average cross-sectional area of every profile containing a dense projection plus two flanking sections. A synaptic vesicle was considered docked if it contacted the plasma membrane. Vesicles that were within 1–4 nm of the plasma membrane that exhibited small tethers to the PM were not scored as docked. The total number of undocked vesicles contacting the dense projection were quantified per profile containing a dense projection.

## Statistical analyses

Statistics was determined using students t-test, one-way ANOVA or two-way ANOVA with Tukey's post-hoc analysis. Error bars were calculated using standard errors of the mean. * signifies $p < 0.05$, **$p < 0.01$, ***$p < 0.001$, ****$p < 0.0001$.

## Acknowledgements

We thank Reiner Bleher for technical assistance with the electron microscopy, Pengpeng Li for assistance constructing the phylogenic tree, Lewie Zeng and Marc Hammarlund for assistance with CRISPR protocols, SoRi Jang, Gabriela Bosque, Lucelenie Rodriguez, Katie Underwood, Gonzalo Tueros and Nathan Cook for help in identifying and characterizing the *cla-1* allele from the forward genetic screens. We would like to thank Jim Rand for discussions of unpublished data. We thank Cori Bargmann, Yishi Jin and Jihong Bai and the *Caenorhabditis* Genetics Center (supported by the National Institutes of Health Office of Research Infrastructure Programs; P40 OD010440) for strains. We thank the Research Center for Minority Institutions program and the Instituto de Neurobiología de la Universidad de Puerto Rico for providing a meeting and brainstorming platforms. DAC-R, ZX and JN were supported by NIH (R01NS076558) and the National Science Foundation (NSF IOS 1353845). PTK and KS were supported by NIH (5R01NS048392) and the Howard Hughes Medical Institute. This work made use of the EPIC facility (NU*ANCE* Center-Northwestern University), which has received support from the MRSEC program (NSF DMR-1121262) at the Materials Research Center; the International Institute for Nanotechnology (IIN); and the State of Illinois, through the IIN.

## Additional information

### Competing interests

Kang Shen: Reviewing editor, *eLife*. The other authors declare that no competing interests exist.

## Funding

| Funder | Grant reference number | Author |
| --- | --- | --- |
| National Institutes of Health | R01NS076558 | Zhao Xuan<br>Jessica Nelson<br>Daniel A Colón-Ramos |
| Howard Hughes Medical Institute | Investigator | Kang Shen<br>Peri T Kurshan |
| National Institutes of Health | 5R01NS048392 | Kang Shen<br>Peri T Kurshan |
| National Science Foundation | NSF IOS 1353845 | Zhao Xuan<br>Jessica Nelson<br>Daniel A Colón-Ramos |

The funders had no role in study design, data collection and interpretation, or the decision to submit the work for publication.

## Author contributions

Zhao Xuan, Conceptualization, Software, Formal analysis, Investigation, Visualization, Methodology, Writing—original draft, Writing—review and editing; Laura Manning, Data curation, Formal analysis, Investigation, Writing—review and editing; Jessica Nelson, Investigation, Writing—review and editing; Janet E Richmond, Conceptualization, Resources, Formal analysis, Supervision, Investigation, Methodology, Writing—review and editing; Daniel A Colón-Ramos, Conceptualization, Resources, Data curation, Supervision, Funding acquisition, Investigation, Writing—review and editing; Kang Shen, Resources, Supervision, Funding acquisition, Writing—review and editing; Peri T Kurshan, Conceptualization, Data curation, Software, Formal analysis, Investigation, Visualization, Methodology, Writing—original draft, Project administration, Writing—review and editing

## Author ORCIDs

Zhao Xuan (iD) http://orcid.org/0000-0002-8254-2887
Laura Manning (iD) http://orcid.org/0000-0003-1597-0600
Kang Shen (iD) http://orcid.org/0000-0003-4059-8249
Peri T Kurshan (iD) http://orcid.org/0000-0001-6267-7103

## Decision letter and Author response

Decision letter https://doi.org/10.7554/eLife.29276.021
Author response https://doi.org/10.7554/eLife.29276.022

# Additional files

## Supplementary files

• Transparent reporting form
DOI: https://doi.org/10.7554/eLife.29276.019

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
