## [Decision Letter]

Thank you for submitting your article "Clarinet (CLA-1), a novel active zone protein required for synaptic vesicle clustering and release" for consideration by *eLife*. Your article has been favorably evaluated by Eve Marder (Senior Editor) and three reviewers, one of whom, Graeme Davis, (Reviewer #1), is a member of our Board of Reviewing Editors. The following individual involved in review of your submission has agreed to reveal their identity: Martin Mueller (Reviewer #2).

In this manuscript, Xuan and colleagues provide the first characterization of a novel nematode active zone protein (clarinet or CLA-1) and investigate its presynaptic role. As the authors mention, there is a fair amount of evolutionary divergence across phyla among active zone proteins that play a structural/organizational role. This is in contrast with the highly-conserved core fusion machinery. To date, worm homologs to bassoon and piccolo have not been described. The authors provide a careful and thorough initial characterization of what appears to be a functional homolog of bassoon and piccolo in mammals and BRP in fly. Using some very nice genomic manipulations as well as imaging, electrophysiology, behavior, and electron microscopy to take full advantage of their model system, the authors conclude that CLA-1 resides in the active zone and may help tether and/or organize SVs as part of normal synaptic function. In addition, CLA-1 may contribute to synapse assembly during development.

Major Comments:

There was an extensive online conversation among the reviewers concerning the manuscript. All reviewers agree that there remains a single major issue that needs additional experimental attention (Comment 1). The additional major concerns are mostly explanations that will help clarify or extend the manuscript. Minor concerns that can be addressed at the author's discretion are listed afterward.

1) A major conclusion of the authors is that CLA-1 is necessary for normal vesicle clustering and release. The EM data underscore the defect in vesicle clustering. However, evoked synaptic transmission following single action potential stimulus is normal. More specifically, the authors need to address how a similar evoked response can be obtained upon single stimulation in *cla-1(S/M/L)* when considering fewer synapses (-30%), a dramatic decrease in the number of vesicles docked to the dense projection and an increase in the number of "stranded" vesicles (Figure 6)? Could an increase in release counteract the decrease in vesicle density/synapse number (see below, 7., 8.)?

Furthermore, *cla-1(S/M/L)* mutants display increased short-term depression and the authors claim that this phenotype is due to a "defect in the number of vesicles that can be readily recruited by depolarization". There remain other interpretations that could explain these data, including changes in vesicle recruitment, replenishment, and release sensors. Indeed, it seems reasonable that a compensatory increase in release could have adjusted release probability to achieve normal release on the first action potential followed by enhanced depression on subsequent action potentials (see Davis and Muller, 2015 for review of presynaptic compensatory mechanisms).

At the heart of this issue is whether CLA-1 actually has a function to control synaptic vesicle release, or whether it has a primary function to control synapse number and vesicle clustering, with the release deficit being a secondary effect. Since this is the first annotation of CLA-1 function, and since it is being proposed as homologous to other central regulators of active zone function, this phenotype needs to be explored in greater detail. Indeed, *cla-1* appears to be similar to Bassoon/Piccolo, which have been implicated in vesicle recovery/replenishment without affecting release after single AP stimulation at most synapses. This is in contrast to a number of mutants, including *brp, rim* or *rbp* that are required for normal release levels upon single AP stimulation.

After considerable discussion, the reviewers suggest two potential avenues that might help resolve this issue, taking into account the unique challenges of the *C. elegans* system for synaptic electrophysiology. One possible approach would be to probe whether *cla-1* is a sensitized genetic background for neurotransmitter release, combining *cla-1* with other gene mutations in key genes involved in synaptic transmission. Ideally, the analysis would be electrophysiology, but if multiple genetic tests were pursued, then behavior could be suitable. This is very standard genetic epistasis in fly and worm. An alternative approach would be to explore the phenotype of enhanced synaptic depression in greater detail. Specifically, could the authors probe recovery from depression and determine if this is altered in *cla-1*? Finally, if the authors have a different approach that would address these issues directly, this would be fine. You may wish to address your response to this in a letter which we would share with the Reviewing Editor and the reviewers for their recommendation.

2) Subsection “Localization of CLA-1 isoforms at the active zone and their role in synaptic vesicle clustering”, first paragraph, the authors estimate an upper bound for the physical extent of CLA-1L. I guess a true upper bound would be if the protein were fully extended N to C with each peptide bond being 0.54 nm long, giving 9000*0.54 = 4.8 microns. Since it appears that much of the protein is disordered, a very different estimate comes from the statistical size of a random coil given approximately by 0.54*(9000)^0.6 ~ 130 nm (see Flory 1949). I didn't find the purely α helical size estimate that useful since it isn't a true upper limit and doesn't describe much of the actual protein secondary structure. Estimates on the order of 200 nm are particularly interesting since they fit nicely with the 3D dense projection reconstructions in Kittelmann. Based on the amount of fluorescence measured in the CLA-1S::mCh or the GFP::CLA-1L imaging, is it possible to estimate the number of copies present at the synapse? I'm just wondering if there is some stoichiometric relationship between the number of SVs contacting a dense projection and the number of CLA-1 copies present. In a *cla-1* heterozygote, would there be a difference?

3) The conclusions that the different isoforms fulfill different roles at the synapse would be strengthened if the authors showed expression of the different isoforms in the different mutants (e.g. by RT-PCR), especially that the M and S isoforms are still expressed normally in the *cla-1(L)* mutant.

4).Authors should use the null allele *cla-1(S/M/L)* for assessing localization of active zone proteins if they want to state that *cla-1* does not affect localization of these proteins at the synapse (subsection “CLA-1 is required in the NSM neuron for synaptic vesicle clustering”, last paragraph).

5) In Figure 1 and Figure 1—figure supplement 1 it looks as if terminal branching of the NSM neuron is altered in the mutant, while this is not the case in the images shown in Figure 1—figure supplement 2. Could the authors comment if this could be part of the phenotype, or whether this is simply a consequence of the representative images chosen?

6) It is not completely clear, especially in panel 6C, how the number of undocked vesicles contacting the DP was quantified.

7) Figure 5: It would be very helpful to show example traces of miniature and evoked EPSCs in this figure. Also, could the authors please show absolute evoked amplitudes in addition to the normalized data, as well as example trains? It would be also helpful to graph the data shown in Figure 5 as a function of time to know the stimulation frequency.

8) It is interesting to compare these data with *syd-2* mutants. In *syd-2*, both minis and evoked EPSCs are down (by 50-60%) and both the total number of SVs and docked SVs are also down by a similar factor (based on Kittelmann 2013). In *cla-1S/M/L* mutants, minis are down by 50% but evoked EPSCs are normal and there is an increase in the number of docked SVs. This seems to imply different relationships between docked SVs, minis, and evoked in the different mutant backgrounds. Since CLA-1S is strongly decreased in *syd-2* mutants (Figure 7), could the more severe phenotype in *syd-2* be closer to a *syd-2;cla-1* double mutant? Did the authors ever look at this double for any genetic interactions? Or is *syd-2* synthetically lethal? This should be discussed.

---

## [Author Response]

Major Comments:There was an extensive online conversation among the reviewers concerning the manuscript. All reviewers agree that there remains a single major issue that needs additional experimental attention (Comment 1). The additional major concerns are mostly explanations that will help clarify or extend the manuscript. Minor concerns that can be addressed at the author's discretion are listed afterward.1) A major conclusion of the authors is that CLA-1 is necessary for normal vesicle clustering and release. The EM data underscore the defect in vesicle clustering. However, evoked synaptic transmission following single action potential stimulus is normal. More specifically, the authors need to address how a similar evoked response can be obtained upon single stimulation in cla-1(S/M/L) when considering fewer synapses (-30%), a dramatic decrease in the number of vesicles docked to the dense projection and an increase in the number of "stranded" vesicles (Figure 6)? Could an increase in release counteract the decrease in vesicle density/synapse number (see below, 7., 8.)?Furthermore, cla-1(S/M/L) mutants display increased short-term depression and the authors claim that this phenotype is due to a "defect in the number of vesicles that can be readily recruited by depolarization". There remain other interpretations that could explain these data, including changes in vesicle recruitment, replenishment, and release sensors. Indeed, it seems reasonable that a compensatory increase in release could have adjusted release probability to achieve normal release on the first action potential followed by enhanced depression on subsequent action potentials (see Davis and Muller, 2015 for review of presynaptic compensatory mechanisms).At the heart of this issue is whether CLA-1 actually has a function to control synaptic vesicle release, or whether it has a primary function to control synapse number and vesicle clustering, with the release deficit being a secondary effect. Since this is the first annotation of CLA-1 function, and since it is being proposed as homologous to other central regulators of active zone function, this phenotype needs to be explored in greater detail. Indeed, cla-1 appears to be similar to Bassoon/Piccolo, which have been implicated in vesicle recovery/replenishment without affecting release after single AP stimulation at most synapses. This is in contrast to a number of mutants, including brp, rim or rbp that are required for normal release levels upon single AP stimulation.After considerable discussion, the reviewers suggest two potential avenues that might help resolve this issue, taking into account the unique challenges of the C. elegans system for synaptic electrophysiology. One possible approach would be to probe whether cla-1 is a sensitized genetic background for neurotransmitter release, combining cla-1 with other gene mutations in key genes involved in synaptic transmission. Ideally, the analysis would be electrophysiology, but if multiple genetic tests were pursued, then behavior could be suitable. This is very standard genetic epistasis in fly and worm. An alternative approach would be to explore the phenotype of enhanced synaptic depression in greater detail. Specifically, could the authors probe recovery from depression and determine if this is altered in cla-1? Finally, if the authors have a different approach that would address these issues directly, this would be fine. You may wish to address your response to this in a letter which we would share with the Reviewing Editor and the reviewers for their recommendation.

Clarinet is required for the structural and morphological development of synapses, as evidenced by the fact that *cla-1* mutants have smaller dense projections and a reduced number of synapses. It is also required for the recruitment and/or tethering of synaptic vesicles to the dense projection. In *cla-1* null mutants, these structural and morphological defects lead to functional consequences, those being a reduction in mini frequency as well as increased depression upon repeated stimulation. However, the response to a single stimulus is normal, suggesting that there may be either redundancies or compensatory mechanisms that can overcome these structural defects. Indeed, we see an increase in the number of docked vesicles within 100nm of the dense projection in *cla-1* mutants (newly quantified in Figure 5), suggesting one potential compensatory mechanism.

The active zone protein UNC-‐10/RIM, which promotes synaptic vesicle release by coupling vesicles to calcium channels, is specifically involved in docking synaptic vesicles at the plasma membrane within 100nm of the dense projection (Weimer et al., 2006). We therefore hypothesized that UNC-‐10/RIM may mediate either a redundant function or a compensatory mechanism responsible for the normal evoked response in *cla-1* mutants, as suggested by the reviewers. To investigate this possibility we recorded evoked responses in *unc-10;cla-1* double mutants. *unc10* single mutants have a pronounced defect in evoked synaptic transmission.

Strikingly, the *unc-10;cla-1* double mutants showed a greatly enhanced defect in evoked release in response to a single stimulus (Figure 6). Importantly, the number of synapses in *unc-10;cla-1* double mutants is not significantly different from *cla-1* mutants alone (Figure 3), indicating that this enhancement of the defect in evoked release cannot be attributed to a synthetic effect on synapse number. These results reveal than in the absence of UNC‐10/RIM, there is indeed a functional consequence of the loss of CLA-1 in the response to a single stimulus, consistent with the hypothesis that redundancy or compensatory mechanisms responsible for the lack of an evoked phenotype in *cla-1* single mutants are mediated by UNC-10/RIM.

2) Subsection “Localization of CLA-1 isoforms at the active zone and their role in synaptic vesicle clustering”, first paragraph, the authors estimate an upper bound for the physical extent of CLA-1L. I guess a true upper bound would be if the protein were fully extended N to C with each peptide bond being 0.54 nm long, giving 9000*0.54 = 4.8 microns. Since it appears that much of the protein is disordered, a very different estimate comes from the statistical size of a random coil given approximately by 0.54*(9000)^0.6 ~ 130 nm (see Flory 1949). I didn't find the purely α helical size estimate that useful since it isn't a true upper limit and doesn't describe much of the actual protein secondary structure. Estimates on the order of 200 nm are particularly interesting since they fit nicely with the 3D dense projection reconstructions in Kittelmann. Based on the amount of fluorescence measured in the CLA-1S::mCh or the GFP::CLA-1L imaging, is it possible to estimate the number of copies present at the synapse? I'm just wondering if there is some stoichiometric relationship between the number of SVs contacting a dense projection and the number of CLA-1 copies present. In a cla-1 heterozygote, would there be a difference?

We have removed this estimate from the manuscript. We have now also included additional data from a new CRISPR line we generated showing that the endogenously tagged N- and C-termini of CLA-1L localize to different regions. The endogenously tagged C-terminus fluorescence localizes precisely to the active zone while the endogenously tagged N-terminus fluorescence is more diffuse (Figure 4), suggesting that the two ends of this protein extend far away from each other.

Unfortunately, it is not possible to estimate protein copy number from fluorescence levels, since it is not clear how we would establish a baseline (e.g. fluorescence from a single molecule).

3) The conclusions that the different isoforms fulfill different roles at the synapse would be strengthened if the authors showed expression of the different isoforms in the different mutants (e.g. by RT-PCR), especially that the M and S isoforms are still expressed normally in the cla-1(L) mutant.

We now include RT-PCR experiments showing that shorter isoforms are still expressed in *cla-1(L)* mutants (Figure 1—figure supplement 1).

4) Authors should use the null allele cla-1(S/M/L) for assessing localization of active zone proteins if they want to state that cla-1 does not affect localization of these proteins at the synapse (subsection “CLA-1 is required in the NSM neuron for synaptic vesicle clustering”, last paragraph).

We now include data showing that CLA-1 is in fact important for the localization of other AZ proteins. Measuring endogenously-labeled SYD-2/Liprin-α in the posterior dorsal nerve cord reveals a significant reduction in the intensity of SYD‐2 puncta (Figure 7). Exogenously expressed SYD‐2 in NSM is also more diffusely localized in *cla1(S/M/L)* mutants (Figure 7—figure supplement 1), mirroring the effects on synaptic vesicles in that neuron.

5) In Figure 1 and Figure 1—figure supplement 1 it looks as if terminal branching of the NSM neuron is altered in the mutant, while this is not the case in the images shown in Figure 1—figure supplement 2. Could the authors comment if this could be part of the phenotype, or whether this is simply a consequence of the representative images chosen?

We did not detect any differences in the terminal branching of NSM neurons in *cla-1* mutant animals and thus have chosen a more representative image for the wild type animal.

6) It is not completely clear, especially in panel 6C, how the number of undocked vesicles contacting the DP was quantified.

This was quantified as the total number of undocked vesicles contacting the dense projection, per profile containing a dense projection. We have clarified in the text.

7) Figure 5: It would be very helpful to show example traces of miniature and evoked EPSCs in this figure. Also, could the authors please show absolute evoked amplitudes in addition to the normalized data, as well as example trains? It would be also helpful to graph the data shown in Figure 5 as a function of time to know the stimulation frequency.

We have added the requested data and information: example traces of endogenous and evoked responses (Figure 6‐B), including the first and last trace from the stimulus train; absolute evoked responses (Figure 6 and Figure 6—figure supplement 1); train data graphed as a function of time (Figure 6 and Figure 6—figure supplement 1).

8) It is interesting to compare these data with syd-2 mutants. In syd-2, both minis and evoked EPSCs are down (by 50-60%) and both the total number of SVs and docked SVs are also down by a similar factor (based on Kittelmann 2013). In cla-1S/M/L mutants, minis are down by 50% but evoked EPSCs are normal and there is an increase in the number of docked SVs. This seems to imply different relationships between docked SVs, minis, and evoked in the different mutant backgrounds. Since CLA-1S is strongly decreased in syd-2 mutants (Figure 7), could the more severe phenotype in syd-2 be closer to a syd-2;cla-1 double mutant? Did the authors ever look at this double for any genetic interactions? Or is syd-2 synthetically lethal? This should be discussed.

Since SYD-2 is required for CLA-1 to localize to the axon (Figure 7), genetically removing *cla-1* in a *syd-2* mutant background would not be expected to exacerbate the *syd-2* phenotype. This is indeed what we found: while *syd-2* mutants show a pronounced defect in synapse assembly, whereby synaptic material form small puncta that are dispersed throughout the axon, the phenotype of *syd-2;cla-1* double mutants was indistinguishable from the *syd-2* single mutants (Figure 7—figure supplement 1). We have added this data to the manuscript.